# Predicting sequence-specific amplification efficiency in multi-template PCR with deep learning

Andreas L. Gimpel [1,6], Bowen Fan[1,2,3,6], Dexiong Chen [2,3,4],
Laetitia O. D. Wölfle [5], Max Horn [2,3], Laetitia Meng-Papaxanthos [2,3],
Philipp L. Antkowiak [1], Wendelin J. Stark[1], Beat Christen [5],
Karsten Borgwardt [2,3,4,6] ✉ & Robert N. Grass [1,6] ✉

Multi-template polymerase chain reaction (PCR) is a critical technique enabling the parallel amplification of diverse DNA molecules, thereby facilitating applications in fields from quantitative molecular biology to DNA data storage. However, non-homogeneous amplification due to sequence-specific amplification efficiencies often results in skewed abundance data, compromising accuracy and sensitivity. In this study, we address amplification efficiency in complex amplicon libraries by employing one-dimensional convolutional neural networks (1D-CNNs) to predict sequence-specific amplification efficiencies, based on sequence information alone. Trained on reliably annotated datasets derived from synthetic DNA pools, these models achieve a high predictive performance (AUROC: 0.88, AUPRC: 0.44), thereby enabling the design of inherently homogeneous amplicon libraries. We further introduce CluMo, a deep learning interpretation framework that identifies specific motifs adjacent to adapter priming sites as closely associated with poor amplification. This insight leads to the elucidation of adapter-mediated self-priming as the major mechanism causing low amplification efficiency, challenging long-standing PCR design assumptions. By addressing the basis for non-homogeneous amplification in multi-template PCR, our deep-learning approach reduces the required sequencing depth to recover 99% of amplicon sequences fourfold, and opens new avenues to improve the efficiency of DNA amplification in fields such as genomics, diagnostics, and synthetic biology.

The technology to amplify DNA via polymerase chain reaction (PCR) is one of the key pillars of molecular diagnostics, enabling many qualitative and quantitative analytical techniques. While classically, PCR is used to amplify a single target sequence, the advent of massive parallel sequencing necessitated simultaneous amplification of many short sequences (i.e., templates) sharing only short terminal adapters. Such multi-template PCR is now fundamental to many routine sequencing preparation workflows and is used across numerous fields, ranging from metabarcoding to DNA data storage[1–3]. However, the simultaneous amplification of many templates poses unique challenges, most

[1]Department of Chemistry and Applied Biosciences, ETH Zurich, Zurich, Switzerland. [2]Department of Biosystems Science and Engineering, ETH Zurich, Basel, Switzerland. [3]Swiss Institute for Bioinformatics (SIB), Lausanne, Switzerland. [4]Department of Machine Learning and Systems Biology, Max Planck Institute of Biochemistry, Martinsried, Germany. [5]Institute of Microbiology, University of Stuttgart, Stuttgart, Germany. [6]These authors contributed equally: Andreas L. Gimpel, Bowen Fan, Karsten Borgwardt, Robert N. Grass. ✉e-mail: borgwardt@biochem.mpg.de; rograss@ethz.ch

importantly imbalanced product-to-template ratios caused by small differences in amplification efficiency between templates[1,4], compromising accuracy and sensitivity of quantitative results[5-8].

While classical PCR with a single, well-defined template is commonly optimized to ensure high amplification efficiency (usually >90%, e.g. via primer design and choice of annealing temperature)[9,10], this is usually infeasible for multi-template PCR. For example, the design of primer and adapter sequences is dictated by sequencing kits for library preparation, and any changes to the amplification conditions might affect templates' amplification efficiencies differently. In addition, each template's amplification efficiency relative to the other templates is most critical for optimal amplification, rather than the absolute amplification efficiency overall. This is because even just a slight amplification disadvantage of one template relative to the others leads to a drastic reduction in its product-to-template ratio[4,5], due to PCR's exponential nature. For example, a template with an amplification efficiency just 5% below the average will be underrepresented by a factor of around two after only 12 PCR cycles, as often used in PCR-based library preparation for Illumina sequencing.

Commonly reported reasons for these small differences in amplification efficiency include degenerate primers, amplicon length, template-product inhibition, amplicon GC content, polymerase choice, temperature profile, and stochastic effects[1,7,11-17]. However, differences in amplification efficiencies are also commonly observed in the field of DNA data storage[11,18-21], in which the properties of the template sequences are well-defined, and sequence information is often deliberately devoid of undesired properties (i.e. extreme GC content, long homopolymers, or secondary structure)[3,18,19,22,23]. This suggests the existence of additional, sequence-specific factors contributing to non-homogeneous amplification efficiencies, independent of most factors previously reported in studies on biological samples[5,11,16].

The relevance of non-homogeneous amplification as a source of bias in multi-template PCR is highlighted by the ongoing efforts for its circumvention. In DNA- and RNA-sequencing, it has prompted the development of unique molecular identifiers[24-26] and PCR-free workflows[27-29] to mitigate or preclude biased abundance data in high-throughput sequencing. In DNA data storage, strategies for DNA immobilization[18,19] and constrained coding[22,23] have emerged to reduce the pool inhomogeneity and sequence dropout caused by deep replication of oligo pools via multi-template PCR. However, despite these efforts to mitigate the bias introduced by non-homogeneous amplification during multi-template PCR, tools for the intrinsic design of efficiently amplifying templates are still missing.

Recent advancements in deep learning have revolutionized DNA sequence analysis, revealing complex characteristics such as DNA-protein interactions[30], effects of non-coding variants[31,32], and chromatin accessibility[33]. However, despite their predictive power and capability to handle large-scale datasets, the 'black-box' nature of deep learning models—such as Convolutional Neural Networks (CNNs)—often limit their ability to elucidate the underlying molecular mechanisms[30,34]. This is in contrast to traditional motif discovery methods[35-39] which offer better interpretability, but are often computationally intensive and lack the discriminative power of deep learning. Therefore, a growing body of research seeks to bridge this gap by extracting interpretable motifs directly from 'black-box' models. DeepBind[30] pioneered this approach by using CNNs for motif discovery, successfully identifying motifs correlated with DNA-protein binding sites by interpreting the convolutional filters in the first layer within the CNNs. Subsequently, DeepLIFT[40] and SHAP[41] further improved feature attribution analysis for general deep learning models, providing nucleotide-level attribution scores based on specific tasks. These local attribution approaches have also been extended by further studies to aggregate for discovery of shared sequence motifs, such as to cluster or aggregate individual attributions into shared

motifs[42], thereby enabling more global model interpretations. Nonetheless, clustering-based global motif identification remains challenging, especially when facing multiple clustering decisions (e.g., cluster number and similarity metrics, managing variable motif lengths[43]).

To overcome these limitations in this work, we employed synthetic oligonucleotide pools to generate large, reliably annotated datasets of sequence-specific amplification efficiencies, and train deep learning models to identify poorly amplifying sequences. Moreover, we present a streamlined motif discovery method—termed Motif Discovery via Attribution and Clustering (CluMo)—to identify specific sequence motifs linked to poor amplification efficiency and quantify their importance. Aided by the model interpretability of CluMo, we can derive a mechanistic understanding of the template-dependent PCR inhibition that causes poor amplification efficiency in multi-template PCR, and facilitate the identification of poorly amplifying templates directly from their sequence.

## Results

### PCR amplification progressively skews coverage distributions

To systematically investigate the inherent, non-homogeneous amplification in multi-template PCR, we first experimentally analyzed the PCR efficiency of individual, synthetic DNA sequences in multi-template PCR reactions. For this, the change in amplicon coverage was tracked for 12,000 random sequences with common, terminal primer binding sites (i.e., truncated Truseq adapters) over 90 PCR cycles using a serial amplification protocol (see Fig. 1). The use of random sequences precluded any bias from enriched sequence motifs present in biological samples (e.g., low-complexity regions). Specifically, six consecutive PCR reactions with 15 cycles each were performed, yielding a sample ready for sequencing at each iteration to quantify precise amplicon composition along the multi-template PCR amplification trajectory (see Supplementary Fig. 1).

During serial amplification, a progressive broadening of the coverage distribution was observed (see Fig. 2a), as previously reported[20,22]. Whereas the overall coverage distribution only changed marginally, a considerable number of amplicon sequences were either severely depleted or even no longer present in the sequencing data (see Fig. 2b). Because the presence of a GC-bias in amplification and sequencing is known[7,11,13,44], the experimental analysis was repeated on a second synthetic oligonucleotide pool in which the random sequences were constrained to 50% GC content (called GCfix). However, the progressive skew of the coverage distribution with increased PCR cycles and the increased fraction of sequences with low coverages was comparable between the GCall and GCfix pools (see Supplementary Fig. 18), suggesting the observed low amplification efficiency of some sequences is not caused by these sequences' GC content.

### Sequence-specific amplification efficiencies for multi-template PCR

In order to translate the observed changes in sequencing coverage to a quantifiable amplification efficiency, a simple fit of the sequencing data to an exponential PCR amplification process[5,12] was performed. This fit uses two parameters per sequence: the initial bias caused by uneven coverage after synthesis[21], and the PCR-induced bias caused by each sequence's individual amplification efficiency ($\epsilon_i$, see Supplementary Fig. 1 for an illustration). The obtained estimates for the initial coverage bias were comparable to experimental data using PCR-free sequencing of oligonucleotide pools[21] (see Fig. 2c, dashed line) and the distributions of amplification efficiencies were comparable across both datasets (see Supplementary Fig. 18). In both datasets the data revealed a small subset of sequences (representing around 2% of the pool) with very poor amplification efficiency (see Fig. 2c, inset). With estimated efficiencies as low as 80% relative to the population mean (equivalent to a halving in relative abundance every 3 cycles), these sequences were often no longer present in the sequencing data after

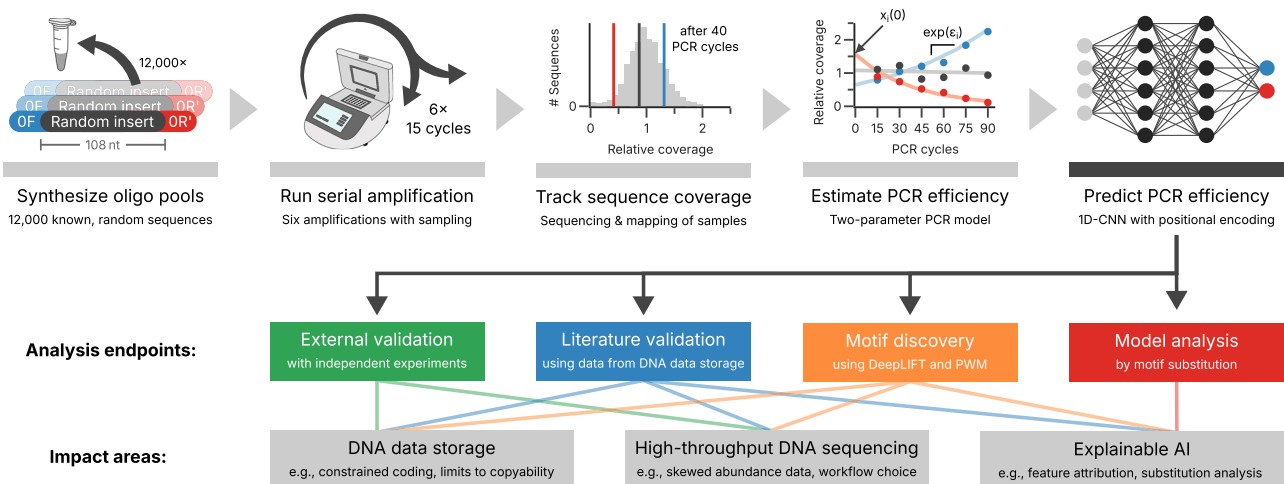

**Fig. 1 | Overview of the workflow and the analysis endpoints.** The workflow starts with the synthesis of randomized oligonucleotide pools with constant adapters (0 F, blue, and 0 R', red), which are consequently amplified serially to generate six samples with differing numbers of PCR cycles (from 15 to 90 cycles). After sequencing, the evolution of each sequence's coverage as a function of cycle number is used to estimate the PCR efficiency in the two-parameter PCR model (see "Methods"). These estimates of the PCR efficiencies are used in the training of an 1D-CNN model for the binary classification of PCR efficiency, see also Supplementary Figs. 1 and 37.

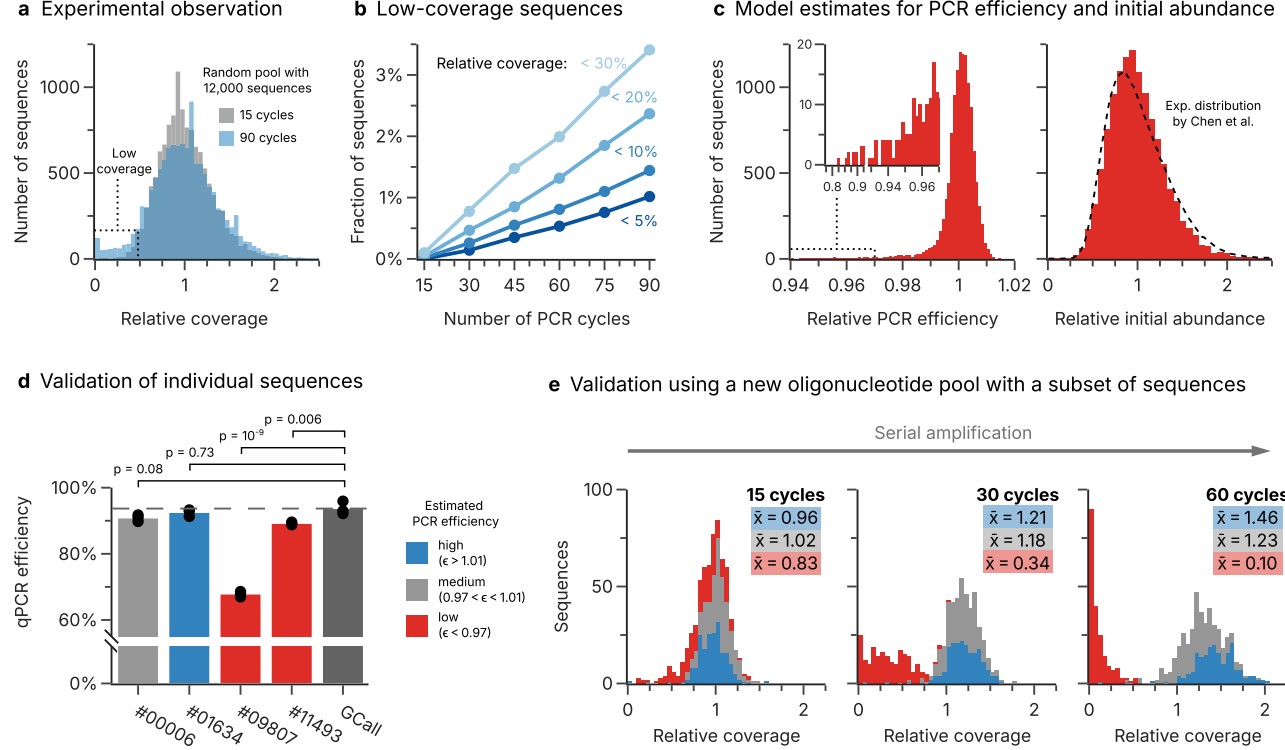

**Fig. 2 | Estimation of sequence-level PCR efficiency and its experimental validation.** **a** Observed normalized coverage distributions of the GCall pool after the first (15 cycles, gray) and the sixth round of serial amplification (90 cycles, blue). **b** Observed fractions of underrepresented sequences in the GCall pool over the course of serial amplification. Low-coverage sequences are further grouped by their relative coverage, from occurring less frequently than 30% (light blue) to lower than 5% (dark blue). **c** Distributions of the relative PCR efficiency (left) and relative initial abundance (right) estimated from the experimental data using a two-parameter fit to exponential PCR amplification (see Methods). The inset for the PCR efficiency shows the subset of sequences with very low amplification efficiency. The dotted line superimposed onto the distribution of relative initial abundance shows the experimentally-determined distribution by Chen et al.[21], using a ready-to-sequence pool. **d** qPCR efficiencies of four individually synthesized, and arbitrarily selected sequences from the GCall pool (#00006 through #11493), and of the GCall pool itself, as measured with qPCR dilution curves. Two of the individual sequences had shown a low amplification efficiency during the serial amplification (#09807 and #11493, red bars). Samples #00006 and #01634 had shown average (gray) and good amplification performance (blue) respectively. Amplification efficiencies were significantly different by one-way ANOVA ($N = 3$ per sample, $F(4, 10) = 252$, $p = 5 \times 10^{-10}$, $\eta^2 = 0.99$), and the results of a post-hoc Tukey's range test are shown above the bars (see Supplementary Tables 2, 3). Black circles show the individual data points. **e** Observed normalized coverage distributions after 15, 30, and 60 cycles of serial amplification (iterations 1, 2, and 4, respectively) of a new pool containing a subset of the sequences present in the GCall and GCfix pools. Sequences were again selected by their estimated PCR efficiencies, and grouped by a high (blue), medium (gray), or low (red) PCR efficiency (see "Methods"). Insets show the mean coverage across all sequences in each category for that experiment. Source data are provided as a Source Data file.

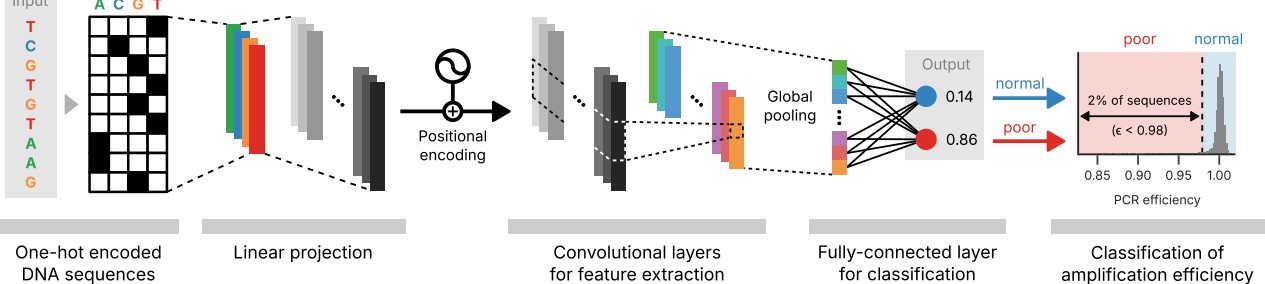

**a** Model architecture and sequence classification

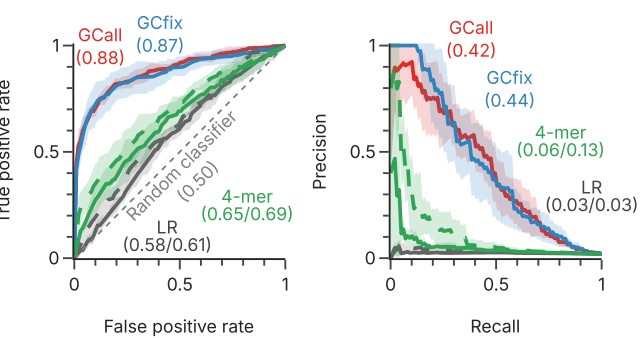

**b** Model performance within datasets

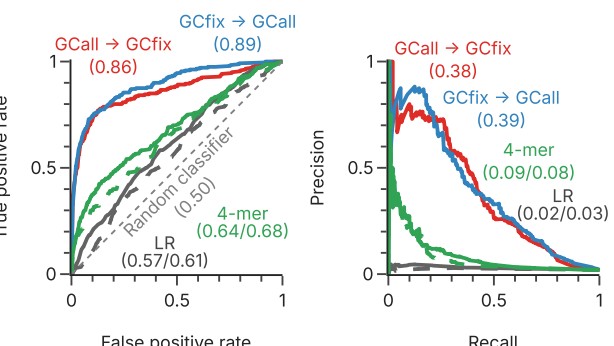

**c** Model performance across datasets

**Fig. 3 | Classification of DNA sequences by estimated amplification efficiency.**
**a** A depiction of the 1D-CNN model with positional encoding to classify sequences' amplification efficiency based on their structural attributes. Batch normalization and Rectified Linear Unit (ReLU) activation are used in-between the convolutional layers. **b** Evaluation of the models' performance within the GCall (red) and GCfix (blue) datasets, presenting mean AUROC (left) and mean AUPRC (right) scores, with the shaded area showing the standard deviation from the five-fold cross-validation. The performance of the baseline models, the LR model, and LightGBM model with 4-mer features, on the datasets is also shown (gray and green, GCall solid, GCfix dashed). **c** Evaluation of the model's performance across datasets, from GCall to GCfix (red) and GCfix to GCall (blue), presenting AUROC (left) and AUPRC (right) scores. The performances of the baseline, the LR model, and LightGBM model with 4-mer features, on the datasets is also shown (gray and green, GCall to GCfix solid, GCfix to GCall dashed). Source data are provided as a Source Data file.

60 cycles. Additional analyses also precluded any potential bias from demultiplexing and mapping after sequencing (see Supplementary Figs. 38, 49–53).

## Poor amplification is reproducible and independent of pool diversity

Two orthogonal experiments were conducted to verify that the sequencing-based quantification of the PCR efficiency is reproducible. For this, the sequences from the two pools (GCall and GCfix, 12,000 sequences each) were categorized by their amplification efficiency (see Fig. 2c and d). In the first experiment, four sequences were arbitrarily selected and their efficiencies were experimentally quantified using dilution curves in single-template qPCR (see Supplementary Fig. 2). Indeed, the sequences with a low amplification efficiency in the sequencing data also had significantly lower amplification efficiencies in qPCR, as shown in Fig. 2d. For the second experiment, a new oligo pool was synthesized, comprising 1000 sequences from the original GCall and GCfix experiments (see Supplementary Fig. 20). Figure 2e shows the evolution of their sequence coverage with increasing PCR cycles during serial amplification, stratified by the sequences' amplification efficiencies. While the differences between sequences with average or high attributed amplification efficiencies are less evident, virtually all sequences with a low attributed amplification efficiency were drastically under-represented even after just 30 PCR cycles, and effectively drowned out completely by cycle number 60 (see Supplementary Fig. 21 for full data). These results show that there are specific amplicon sequences which significantly and reproducibly amplify less efficiently than the remaining sequences in the pool, and that this does not depend on the composition of the pool, but rather on their sequence.

## Positional sequence information is critical to predict poor amplification with deep learning

In order to understand why the amplification of a fraction of sequences in multi-template PCR is hampered, we focused on the worst-performing 2% of sequences from the GCfix and GCall experiments (see Fig. 3a and "Methods", other thresholds in Supplementary Figs. 23–25). Motivated by literature on GC-induced PCR bias[11,13,16], we first attempted to explain the data using a Lasso regularized logistic regression (LR) model, with GC content and base frequencies as features. However, the regression model performance is poor (see Fig. 3b, c, gray lines), with a prediction accuracy close to a random classifier. This indicates that poor amplification of some sequences cannot be explained by the base composition or GC content alone, which is also confirmed by the similarity between the datasets with variable and fixed GC content (GCall and GCfix, see Supplementary Fig. 18). As a second baseline, k-mer based models with positional information were implemented ($k \in [3, 5]$, see Supplementary Note 2 and Supplementary Fig. 27). While the best-performing 4-mer model improved slightly upon the regression model (see Fig. 3b, c, green lines), its predictive performance still remains poor.

As an alternative to these baseline models, three deep-learning models were trained on the datasets: a RNN, a 1D-CNN, and a 1D-CNN with positional encoding. While all deep-learning models outperformed the baseline models considerably (see Supplementary Fig. 22), the 1D-CNN with positional encoding—outlined in Fig. 3a was selected for further investigation due to its superior predictive power. When evaluated within our datasets (GCall and GCfix), the 1D-CNN models with positional encoding show an average AUROC (Area Under the Receiver Operating Characteristic curve) of 0.88 and 0.87, and an average AUPRC (Area Under the Precision-Recall Curve) of 0.42 and 0.44, respectively

**a** Workflow for motif discovery

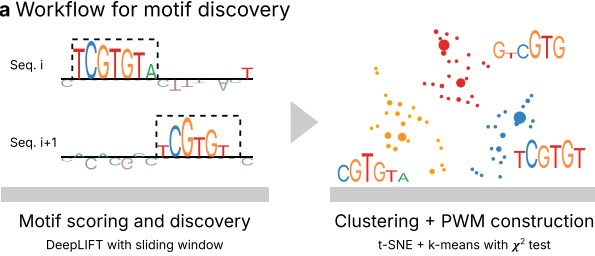

Motif scoring and discovery
DeepLIFT with sliding window

Clustering + PWM construction
t-SNE + k-means with $\chi^2$ test

**b** Most significant motifs are similar between models

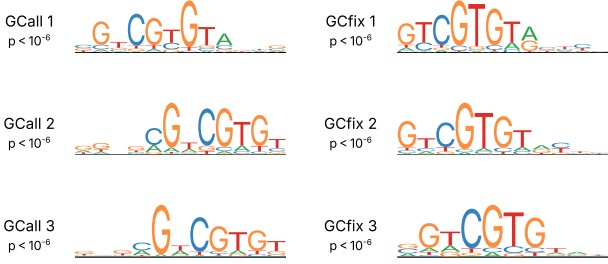

**c** Positional bias in the motif frequency

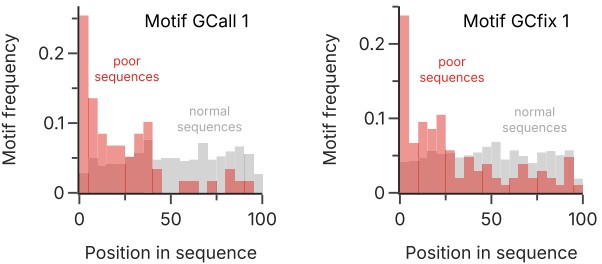

**d** Workflow for testing motif relevance by substitution

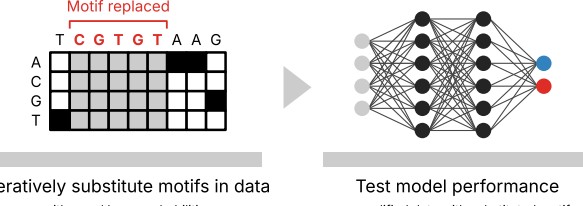

Iteratively substitute motifs in data
with equal base probabilities

Test model performance
on modified data with substituted motifs

**e** Motifs explain majority of model performance

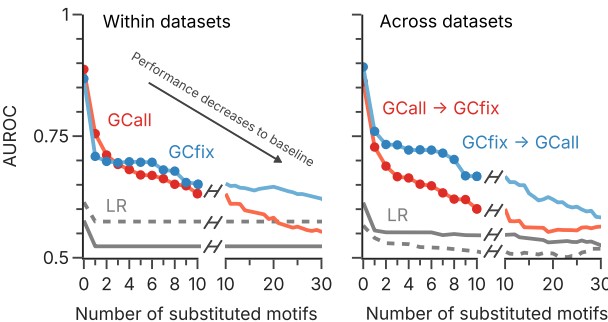

**f** Mechanism for inhibition by motifs

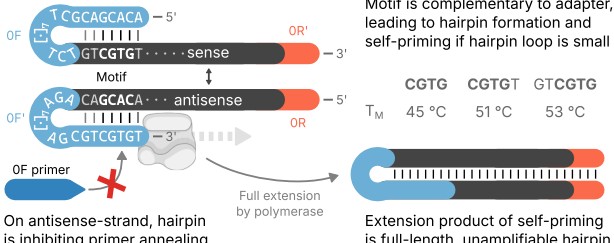

**Fig. 4 | Discovering motifs affecting amplification efficiency and testing their relevance. a** Workflow for discovering motifs with CluMo, based on motif extraction with DeepLIFT[40], t-distributed stochastic neighbor embedding (t-SNE), and k-means clustering. The significance of the resulting position weight matrices (PWM) are assessed with a chi-squared test. **b** Most significant motifs identified for the GCall (left column) and GCfix datasets (right). The significance of the chi-squared test of each PWM is also shown. **c** Positional bias in the occurrence of the two motifs GCall 1 and GCfix 1 (see Panel **b**). The motifs in the poorly amplifying sequences (red) are more frequent at the beginning of the sequence, whereas there is no bias in normal sequences (gray). More data is shown in Supplementary Fig. 12. **d** Workflow for validation the discovered motifs in the trained model by iteratively replacing the motifs in the test data (ordered by *p*-values) without further retraining. The decrease in predictive power of the model upon motif replacement is correlated to that motif's relevance to the model output. **e** Evolution of model

AUROC as a function of the number of substituted motifs in the test data, either using internal validation (left) or testing across datasets (right). The models trained on GCall (red) and GCfix (blue) approach the performance of the baseline LR model (gray; GCall solid, GCfix dashed) as the number of substituted motifs increases. An evaluation using AUPRC as model performance metric is shown in Supplementary Fig. 4. **f** Hypothesized inhibition mechanism explaining the amplification disadvantage conveyed by the CGTG (sub-)motif. The motif is complementary to the 5'-adapter present on all oligos, thereby enabling hairpin formation and self-priming. This inhibits primer annealing and leads to the formation of full-length hairpins which cannot be amplified further. An identical mechanism also enables hairpin formation at the 3-adapter, see Supplementary Fig. 17. Melting temperatures of the hairpins involving different motifs were calculated with mfold[45]. Panel **f** partially created in BioRender. Gimpel, A. (2025) https://BioRender.com/19eu0ya. Source data are provided as a Source Data file.

(see Fig. 3b) using fivefold cross-validation. Further variations of the model architecture, such as oversampling or regression, did not further improve model performance (see Supplementary Note 2 and Supplementary Figs. 28, 29). The evaluation across the two datasets shows the generalizability of the models, achieving largely identical performance when evaluated on each other's data (see Fig. 3c). Importantly, the performance of the 1D-CNN is greatly affected by the introduction of positional encoding, indicating that poor amplification efficiency is mainly attributable to position-specific features within the template sequence.

### Discovery and validation of positional motifs associated with low PCR efficiency

The analysis above shows that 1D-CNN models perform well in identifying sequences with poor amplification efficiency from their

sequence information alone. However, the black-box nature of the 1D-CNN models does not allow an interpretation of the sequence features responsible for this performance. To remedy this, we developed a motif discovery approach called CluMo to elucidate sub-sequence features, based on DeepLIFT[40], which was extended using k-mer analysis and clustering to interpret the trained model (see Fig. 4a).

Applying CluMo to the GCall and GCfix datasets identified several positional motifs strongly associated with poor amplification efficiency (see Fig. 4b and Supplementary Fig. 12). Interestingly, most identified motifs include a common *CGTG* subsequence, usually flanked by similar nucleotides with lower attribution scores. Further analysis reveals that these motifs exhibit a marked propensity to occur at the beginning of poorly amplified sequences (i.e. adjacent to the primer binding site), a trend that is not observed in sequences with normal amplification efficiency (see Fig. 4c). The presence and

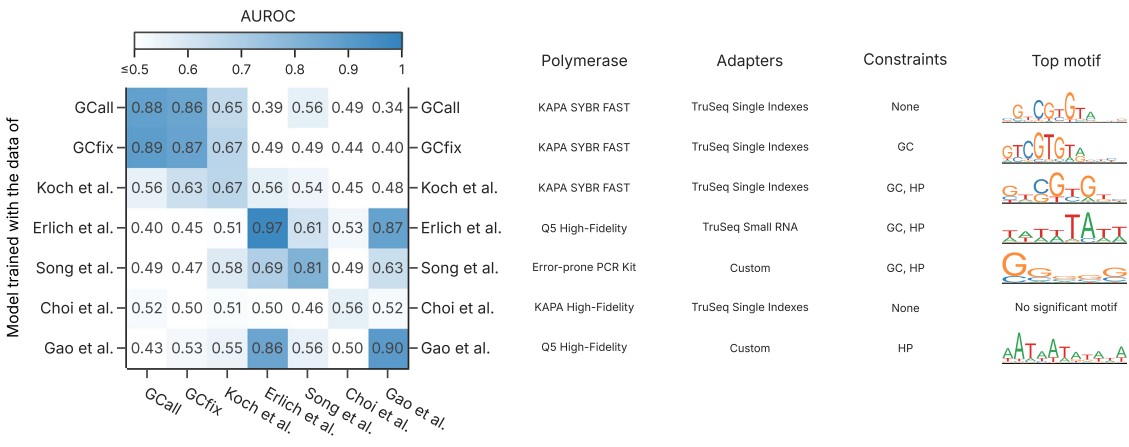

**Fig. 5 | Assessing model performance across literature datasets.** Heatmap of the area under the receiver-operating characteristic (AUROC) metric for the models trained and tested on the different literature datasets. Additional information on the choice of polymerase and amplification adapters is provided, together with constraints used during sequence design and the highest-ranked motif identified by CluMo. Additional information on the literature datasets is provided in Supplementary Table 7, and the corresponding heatmap using the area under the precision-recall curve (AUPRC) is given as Supplementary Fig. 5. Source data are provided as a Source Data file.

positional bias of this (sub-)motif was further validated by k-mer analysis (see Supplementary Figs. 6 and 7).

In order to assess the importance of motifs for model performance, we iteratively replaced motifs, ranked by their *p* values, in the test data with elements having equal base probabilities (see Fig. 4d). Evaluating the models' classification performance after the substitution of each motif highlighted a pronounced decline toward baseline performance, as quantified by a considerable drop in AUROC (see Fig. 4d, e, and Supplementary Fig. 4 for AUPRC). This degradation highlights the critical role these motifs play in the predictive power of the model.

**Adapter-mediated self-priming impedes template amplification**
The interpretability of the trained 1D-CNNs afforded by CluMo provided two important insights into the PCR-induced bias in the GCall and GCfix datasets: the role of *CGTG*-based sequence motifs, and their proximity to adapter sequences. Motivated by the latter, we looked for potential interactions between the terminal adapter and the uncovered sequence motifs. Indeed, all motifs identified by CluMo share short complementarities to the 5'-end of the 5'-adapter which support the formation of hairpins (see Fig. 4f). Importantly, a similar hairpin can also form with the complementary template strand, involving the 3'-end of the 3'-adapter. This hairpin structure inhibits primer annealing and enables self-priming, both of which will decrease amplification efficiency.

To investigate the plausibility of the hypothesized amplicon inhibition through self-priming, we first considered thermodynamic stability of the proposed hairpin during the PCR's annealing step (i.e. at 54 °C with 50 mM Na⁺) by hybridization prediction[45]. Despite the small size of its stem, hairpin formation was competitive to primer annealing, with predicted melting temperatures of 45–53 °C (see Fig. 4f and Supplementary Fig. 15c). In line with literature data[46,47], extending the hairpin loop by moving the motif in the 3'-direction decreased its thermodynamic stability, and thereby diminished its expected impact on amplification efficiency (see Supplementary Fig. 15a). This explains the observed positional enrichment of motifs towards the 5'-end in poorly amplifying sequences (see Fig. 4c). Consistent with a self-priming induced inhibition mechanism, specific motifs with complementarity to the 3'-end of the opposite adapter were also highly enriched within poorly amplifying sequences (see Supplementary Fig. 16).

To further validate whether self-priming by motif-adapter hairpins affects amplification efficiency, we determined amplification rates for the four individual sequences presented in Fig. 2d using degenerate primers and qPCR (see Supplementary Fig. 3). These primers extended the previously used primers with four degenerate A/T nucleotides at each 5'-end, in order to introduce complementary A/T tails into template sequences during amplification. Due to these tails' inability to hybridize with the template - thereby impeding extension by polymerases - we expect self-priming to be suppressed and amplification efficiency to be restored to normal levels (see Supplementary Fig. 17b for illustration). Indeed, qPCR with degenerate primers showed full recovery of amplification efficiency for the previously poorly amplifying sequence with a motif (#09807, from 68 ± 2% to 81 ± 5%), whereas all other sequences' efficiencies decreased slightly (see Supplementary Fig. 8).

To assess the generalization of the self-priming mechanism to other adapters, the amplification efficiencies of sequences with different adapters were also quantified by qPCR. For this, an amplicon and corresponding amplification adapters were picked automatically by Primer3Plus[48] from a long, random sequence (see Supplementary Note 5 and Supplementary Fig. 31). Expectedly, this sequence without any motif amplified efficiently (amplification efficiency of 94.0 ± 1.9%). Upon introduction of 6 nt motifs complementary to either the new 5'- or 3'-adapter into this sequence, the amplification efficiency dropped significantly (5'-motif: 78.8 ± 2.1%, 3'-motif: 81 ± 4%, see Supplementary Fig. 31). This effect is in line with the proposed mechanism of self-priming by motif-adapter hairpins, demonstrating its general validity and extension to other primer sets. Further investigations using melting curve analysis and extension without primers further validated the self-priming mechanism, confirmed the presence of extended hairpins, and ruled out any dependence on initial template concentration (see Supplementary Figs. 32–36).

**Generalization and benchmarking to literature datasets**
To assess the generalizability of our model and benchmark the existence of similar motif-dependent biases in other workflows, we investigated the performance of the 1D-CNN model on a range of literature datasets from the DNA data storage community[18,19,22,49,50]. Contrary to literature data from biological fields, all of these datasets include sequencing data of deep copies (i.e., after at least 100 PCR cycles), and use well-defined synthetic DNA sequences. However, these datasets differ in their experimental conditions (e.g., choice of polymerase and adapter, see Supplementary Table 7). The performance of the 1D-CNN models trained and evaluated across all datasets is shown

in Fig. 5 (for AUPRC, see Supplementary Fig. 5). As expected, all models perform best internally (diagonal in Fig. 5), but performance on other datasets is generally poor. Notable exceptions are two groups of datasets whose models exhibit some transferability between them: GCall, GCfix, and Koch et al.[49] (mean AUROC/AUPRC: 0.71/0.17), as well as Erlich et al.[22] and Gao et al.[19] (0.87/0.17).

For both groups of datasets exhibiting model transferability, the use of common experimental conditions (i.e., polymerase and/or adapters, see Fig. 5 and Supplementary Table 7) suggests a workflow-dependent nature of poor amplification efficiency. Indeed, using CluMo to investigate relevant submotifs, the previously discussed *CGTG* motif also arises in the model of Koch et al.[49], with a similarly strong positional preference towards the 5′-adapter (see Supplementary Fig. 12). The presence of this motif in the data of Koch et al.[49], therefore underscores the predictive power of the models trained on GCall or GCfix, and illustrates the presence of the poor amplification efficiency hypothesized to stem from adapter-mediated self-priming in literature datasets.

In contrast, the transferability between models trained on Erlich et al.[22] and Gao et al.[19] likely stems from the Q5 polymerase used in both studies. Analysis by CluMo reveals a strong association between low amplification efficiency and A/T-rich regions at either end of the sequence for these two models (see Supplementary Figs. 12 and 13). This low GC-content, particularly localized to within a few nucleotides downstream of the adapter, is known to affect amplification efficiency due to next-base effects[51,52] and the GC-preferences of Q5 polymerase[17] and processivity-enhancing dsDNA binding domains[53,54]. A similar argument might also explain the GC-bias extracted by CluMo in the data of Song et al.[50], using an error-prone polymerase for amplification[13].

### External validation of model performance and motif effects
To validate both the performance of the trained 1D-CNN models and the inhibitory effects of the identified motifs, another oligo pool was prepared and amplified in an external laboratory. This pool, containing 10,000 random sequences and 2000 sequences with deliberately inserted motifs, also underwent serial amplification, once with the experimental conditions of GCall and GCfix (KAPA SYBR FAST, 54 °C annealing), and once with an altered protocol based on Erlich et al.[22] (i.e., Q5 HiFi polymerase and 60 °C annealing, see Fig. 6a and "Methods").

Focussing on the experimental conditions of GCall and GCfix, we first evaluated the performance of the models trained with the GCall and GCfix data on the external validation dataset, using only the sequences with random inserts. Reassuringly, these models reached virtually identical predictive power as previously found in internal validation (AUROC 0.8, AUPRC 0.3, see left half of Fig. 6b), underlining the robustness of the models and the reproducibility of poor amplification efficiency across laboratories. Turning to the sequences with inserted motifs, we quantified the amplification disadvantage conveyed by a deliberately inserted motif with the change in PCR efficiency between an individual sequence and its copy with the inserted motif (replacing existing nucleotides, see Fig. 6c). Across the ten motifs deliberately inserted into forty randomly selected sequences at five different positions each (see Fig. 6a), we observed profound inhibitory effects on PCR efficiency depending on motif and position (see Fig. 6c, left, and Supplementary Fig. 9). In line with the hypothesized mechanism of hairpin-mediated self-priming shown in Fig. 4f, the effect of *CGTG*-derived motifs were most prominent at or close to the 5′- and 3′-ends. At the extreme, the motif *TCGTGT* inserted directly downstream of the 5′-adapter led to a mean decrease in PCR efficiency of 4.8 ± 2.4%, equivalent to a halving of the relative abundance approximately every 14 cycles.

Focussing next on the experimental conditions altered to reflect the polymerase and annealing temperature chosen by Erlich et al.[22], the aforementioned workflow-dependence of the poor amplification efficiency becomes obvious. Assessing the performance of the model trained with the literature data of Erlich et al.[22] on the validation dataset revealed a very poor predictive power (AUROC 0.43, AUPRC 0.02; see the right half panel of Fig. 6b). Interestingly, a residual inhibitory effect on amplification efficiency is still conveyed by the motifs previously identified to partake in adapter-mediated self-priming (see Fig. 6c, upper right, and Supplementary Figs. 10 and 11). This is not surprising, given that a key difference between the experimental conditions by Erlich et al.[22] and those used here is the choice of adapters, which appears to supersede or otherwise interact with the poor, GC-based amplification efficiency conveyed by the Q5 polymerase.

To highlight the impact of the trained 1D-CNN model on amplicon sequencing and DNA data storage, we compared the performance of the 1D-CNN with the state-of-the-art, i.e., sequence constraints on GC content, homopolymer lengths, and free energy[22,23]. As shown in Fig. 6d, the 1D-CNN-based filtering completely prevents PCR-induced bias, thereby reducing the number of low-coverage sequences after deep amplification. As a result, the trained 1D-CNN reduces the required sequencing depth to achieve 99% sequence recovery fourfold (see Fig. 6e).

Taken together, the results of the external validation strongly support the motif effects identified by CluMo and the robustness of the trained models. At the same time, the poor performance of the models in the external validation using a mixed condition of polymerase, annealing temperature, and adapters illustrates the strong workflow dependence of the identified poor amplification efficiency in multi-template PCR. Nonetheless, the external validation proved that, as long as all amplification conditions are kept identical, the 1D-CNN models trained on the GCall/GCfix datasets perform well on data generated by an external laboratory and exhibit the same motif-dependent amplification inhibition that we show to be caused by adapter-mediated self-priming.

## Discussion
Leveraging the versatility and flexibility of synthetic oligonucleotide pools, this work identifies multiple motif-dependent biases in multi-template PCR using explainable machine learning. For this, an experimental method was established to reproducibly annotate synthetic sequences with their PCR efficiencies, and a framework of motif discovery and analysis—called CluMo—was developed to interpret deep learning models trained on this data. With these tools, we identify poor amplification efficiency affecting about 2% of random sequences in our datasets, and exploit the insights generated by explainable machine learning to pinpoint short position-dependent sequence motifs conveying an amplification disadvantage by adapter-mediated self-priming (shown in Fig. 4f).

Compared to the existing body of research[5,7,8,12,16,17] on PCR-induced bias in multi-template PCR, our use of synthetic oligonucleotide pools overcomes key bottlenecks which previously precluded exploitation of deep learning, namely dataset size and annotation quality. We found 1D-CNN models with positional encoding can accurately identify poorly amplifying sequences in our datasets (AUROC > 0.8, AUPRC > 0.4), and demonstrated their reliability with data generated in an external laboratory. Importantly, using deep learning enabled a hypothesis-free investigation into amplification efficiencies during multi-template PCR, without relying on pre-defined predictors such as GC content or k-mers. As a result, testing model generalization with literature datasets exposed the workflow-dependence of poor amplification efficiency while interpreting model performance with CluMo revealed adapter- and polymerase-specific motifs as its source. The learning from this analysis also applies to classical single-template PCR, where short-motif self-priming should also be accounted for to optimize amplification efficiency.

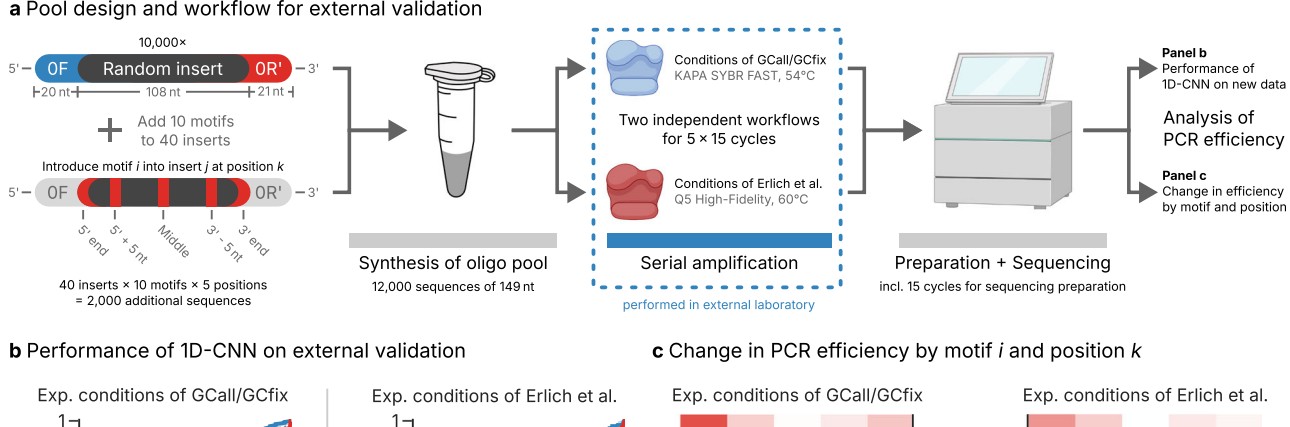

**a** Pool design and workflow for external validation

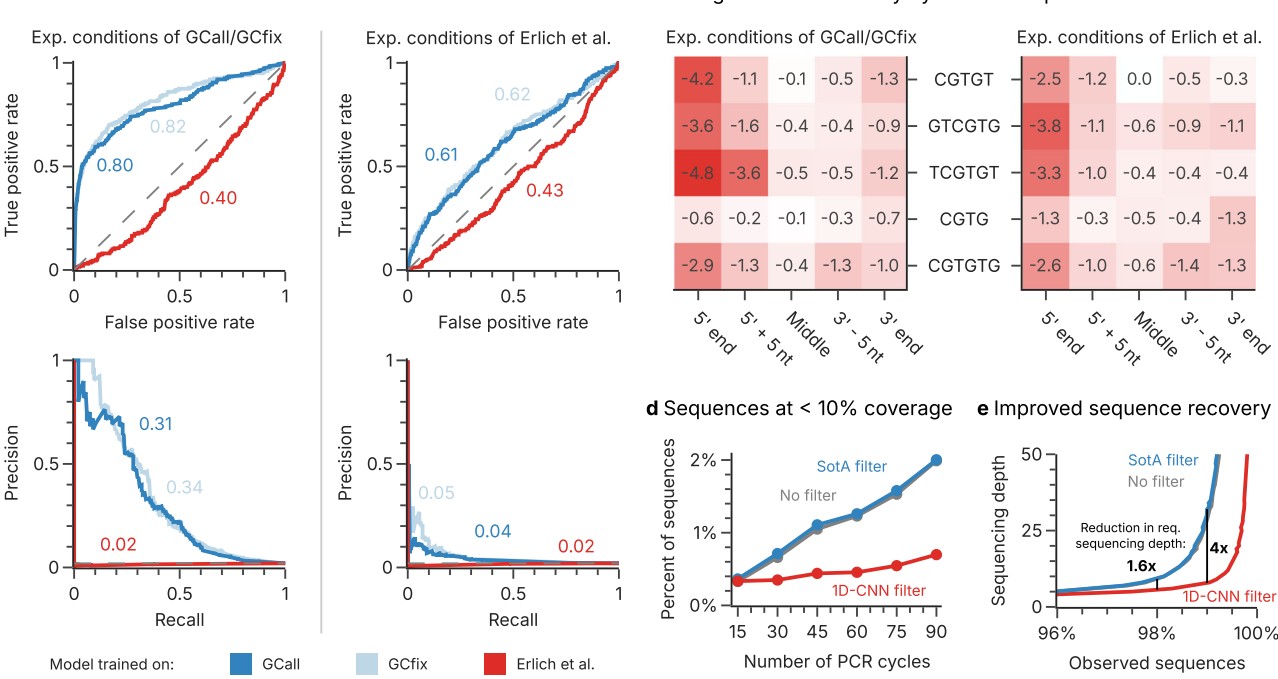

**Fig. 6 | Externally validating model performance and motif discovery with a motif-enriched oligonucleotide pool. a** For external validation, a new oligonucleotide pool consisting of 10,000 random sequences and an additional 2000 sequences with deliberately inserted motifs was used. This pool underwent serial amplification in two independent, parallel workflows, which differed in the polymerase and the temperature profile used during PCR. The resulting sequencing data and estimated PCR efficiencies were used for external validation of model performance and motif discovery. **b** Performance metrics of the 1D-CNN models trained on the GCall (dark blue), GCfix (light blue), or Erlich et al. (red) datasets when tested on the data of the external validation experiments. The area under the receiver operating characteristic (AUROC, left) and the area under the precision-recall curve (AUPRC, right) are shown in the plot. The performance metrics are shown both for the datasets created with the conditions of GCall/GCfix (left column) and Erlich et al. (right column). **c** The mean change in amplification efficiency ($\Delta\epsilon$) across all sequences with a specific, deliberately inserted motif (vertical axis) at a specific position (horizontal axis) compared to the sequence without inserted

motif. In both workflows (left column: conditions of GCall/GCfix, right column: Erlich et al.), the presence of the motifs identified during motif discovery for the GCall/GCfix models lead to a considerable decrease in amplification efficiency if present at, or close to, the 5' end of the sequence. **d** Comparison of the effects of filtering on the incidence of sequences with less than 10% coverage upon amplification of the validation pool. Compared to all sequences (No filter, gray), the state-of-the-art filtering (SotA filter, blue) includes constraints on GC-content, homopolymer length, and free energy (see "Methods"). Filtering using the trained 1D-CNN (1D-CNN filter, red) was implemented by removing the same number of sequences as the SotA filter based on the 1D-CNN's sequence score. **e** Comparison of the effects of filtering on the minimum sequencing depth required to achieve high sequence coverage after amplification of the validation pool for 90 cycles. Filters are identical to Panel d). Panel a) partially created in BioRender. Gimpel, A. (2025) https://BioRender.com/19eu0ya. Source data are provided as a Source Data file.

While hairpin formation is a common design consideration for primers[10], its ability to enable self-priming of templates by short-ranged, motif-driven interactions with PCR adapters has only been reported for reverse transcription[55–57] (i.e., at 37 °C). However, our thermodynamic analysis and qPCR-based experiments suggest that these motif-adapter hairpins convey an amplification disadvantage in multi-template PCR by adapter-mediated self-priming, despite PCR's higher temperatures (54–60 °C) and the motifs' short lengths (4–6 nt).

CluMo extends beyond the specific application in this work and provides a general framework for interpreting neural networks that

process sequence data. Building on local attribution methods such as DeepLIFT[40] and SHAP[41], it systematically clusters high-impact sub-sequences across multiple sequences of any length, thereby uncovering globally significant motifs while quantifying their statistical enrichment and potential interactions. In doing so, CluMo complements prior motif-discovery pipelines that aggregate local attribution scores for example, TFMoDisco[42], but remains model-agnostic. Unlike classic convolution-based visualization methods such as DeepBind[30], DeepSEA[31], or CKN[58], CluMo can be flexibly applied to any neural architecture (CNN, RNN, transformer and etc.), making it broadly useful for diverse sequence analysis tasks.

We envision accurate predictors for amplification efficiency and the tools required to develop them, such as those presented in this work, will find applications in fields such as genomics, diagnostics, and synthetic biology. Besides their use in constrained coding for DNA data storage[22,23], research relying on PCR-based workflows for high-throughput sequencing—such as metabarcoding or RNA sequencing—will benefit from the ability to identify and correct for PCR-induced biases in quantitative sequencing data.

## Methods

### Design of oligonucleotide pools

All oligonucleotide pools used in the experiments of this study were purchased from Twist Biosciences (Piscataway, NJ, United States) and designed with a fixed length of 149 nt. In all oligo pools, each design sequence contained a unique subsequence of 108 nt flanked by constant primer adapters (0 F, 20 nt and 0 R, 21 nt, see Supplementary Table 6), thereby following established protocols[3] for sequencing preparation with the Illumina Truseq protocol.

The two oligonucleotide pools comprising 12,000 sequences—either with (GCfix) or without (GCall) a fixed GC content of 50%—contained fully randomly generated subsequences. These subsequences were generated by iteratively choosing nucleobases at equal probabilities using the Mersenne Twister pseudorandom number generator implemented in Python's random library. These pools and their sequencing data have previously been used to estimate error rates and biases in the DNA data storage workflow[20].

The test pool used for assessing the reproducibility of the amplification bias comprised 1000 sequences which were selected in equal parts from the sequences in the GCall and GCfix pools. The selection of sequences was based on the parameter estimates of two fitting procedures: one set of parameter estimates using the simple exponential PCR equation of Eq.( 1), and a second set of parameter estimates using an extended fitting procedure including a parameter for sequencing efficiency and an estimate of parameter variability (not discussed in main text, see Supplementary Note 3). For both fitting procedures, the best- and worst-performing 100 sequences of the GCall and GCfix pools were selected. Due to a small intersection of these two sets, a total of 400 sequences were selected from the simple parameter-fitting procedure and only 346 sequences from the extended parameter-fitting procedure. The remaining 254 sequences were randomly selected from the sets of sequences from both pools. As the estimates of the simple parameter-fitting procedure matched the experimental results at least as well as the extended parameter-fitting procedure (see Supplementary Fig. 20 and Supplementary Note 3), the 346 sequences of the latter were not considered further. As a result, the analysis in Fig. 2e only shows the 654 individual sequences selected from the simple parameter-fitting procedure and the random sampling.

The validation pool used for external validation of the machine learning model and the effect of motifs comprised a total of 12,000 sequences. Of those, 10,000 were fully randomly generated without any constraint on GC content, as before. For the remaining 2000 sequences, we selected the five most significant motifs inferred from the models trained on the GCall/GCfix datasets or the literature dataset by Erlich et al.[22] and created additional sequences with these motifs. To do so, 40 sequences of the randomly generated subset that did not already contain any of the motifs were selected, and each motif was inserted into each sequence once at the start, the end, the middle, 5 nt from the start, and 5 nt from the end of the sequence, replacing the nucleotides present there. This resulted in 50 additional sequences for each of the 40 sequences selected from the subset, for a total of 2000 additional non-random sequences. Due to the overlapping nature of some motifs (e.g., motif CGTG is contained in motif CGTGT), the resulting set of sequences contained duplicates (e.g,. if the sequence randomly featured T at the position following the inserted motif

CGTG). In the analysis of the sequencing data, these sequences were always associated with the longer motif (e.g. to CGTGT in the example above) to isolate the effects of short motifs as best as possible.

### Serial amplification of pools

All oligonucleotide pools were dissolved to 10 ng µL$^{-1}$ in ultrapure water (Barnstead MicroPure UV water purification system), vortexed, and a 500x dilution in ultrapure water was used as starting point for serial amplification. All experiments except the second serial amplification of the validation pool used KAPA SYBR FAST polymerase master mix (SFLCKB, Roche, Basel, Switzerland) for amplification, employing a temperature profile with an initial denaturation at 95 °C for 3 min, followed by 15 cycles at 95 °C for 15 s, 54 °C for 30 s, and 72 °C for 30 s, using a Roche LightCycler 480 II. Finally, a final extension at 72 °C for 3 min was performed. The second serial amplification of the validation pool used Q5 Hot Start High-Fidelity master mix (M0494, New England Biolabs, Ipswich, MA, United States) instead, employing a temperature profile with an initial denaturation at 98 °C for 30 s, followed by 15 cycles at 98 °C for 10 s, 60 °C for 30 s, and 72 °C for 30 s[22]. Finally, a final extension at 72 °C for 5 min was performed. For each amplification, 5 µL of sample were mixed with 10 µL of 2x master mix, 3 µL of ultrapure water, and 1 µL each of the forward and reverse primer at 10 µM (Microsynth AG, Balgach, Switzerland). An overview of the primer sequences used in our experiments is given in Supplementary Table 6.

The serial amplification of oligonucleotide pools followed an iterative protocol, as previously described[20], to prevent resource exhaustion during PCR. Specifically, each iteration started by diluting 1 µL of the sample from the previous iteration by a factor of 3800x in ultrapure water (or 7600x, if the sample had approached the plateau phase after 15 cycles in the previous iteration). Then, the sample was amplified for 15 cycles in two wells: once using the standard primers (0 F/0 R), and once using primers with an overhang containing indexed sequencing adapters (2FUF/2RIF), see Supplementary Table 6. The PCR product with sequencing adapters was then stored at −20 °C, whereas the PCR product with the standard primers was directly used for the next iteration.

In the case of the validation pool, the amplifications with the standard primers (0 F/0 R) were performed at the Institute of Microbiology of the University of Stuttgart. There, a Barnstead NANOpure II water purification system and a BioRad C1000 Touch thermocycler were used, while the above-mentioned qPCR kits were supplied by the Functional Materials Laboratory of ETH Zurich. The amplification with sequencing adapters (2FUF/2RIF) of the validation pool, as well all other serial amplification experiments and sequencing were performed at the Functional Materials Laboratory of ETH Zurich.

### Sequencing and data preprocessing

The PCR product with indexed sequencing adapters was purified by excision of the appropriate band on an agarose gel (E-Gel EX Agarose Gels 2%, G401002, Invitrogen, Waltham, MA, United States) with a 50 bp DNA ladder (10416014, Invitrogen, Waltham, MA, United States), and subsequent spin-column purification (ZymoClean Gel DNA Recovery Kit, D4007, ZymoResearch, Irvine, CA, USA). All samples were quantified by fluorescence (Qubit dsDNA HS Kit, Q32851, Invitrogen, Waltham, MA, United States) prior to dilution to 1 nM with ultrapure water. Multiple samples were then pooled, further diluted to 50 pM, and finally sequenced on the iSeq 100 (iSeq 100 i1 Reagent v2, 20031371, Illumina, San Diego, CA, USA) with 150 bp paired reads.

The demultiplexed sequencing data was post-processed by adapter trimming and read mapping using BBMap[59] (v39.01) against the pool's reference sequences. Reference sequences whose reads occurred in fewer than two sequencing runs across the dataset were removed from the data. The read counts for all remaining reference sequences,

normalized by the mean number of reads per reference sequence in the dataset, were used as coverage distributions for further analysis.

## Parameter estimation from coverage distributions

To estimate the synthesis bias $x_i(0)$ and the relative amplification efficiency $\epsilon_i$ of each reference sequence $i$ in a set of serial amplification experiments, we model the evolution of the relative sequence coverage $x_i(c_j)$ after $c_j$ cycles as[5,12,20]

$$x_i(c_j) = x_i(0) \cdot \epsilon_i^{c_j}. \tag{1}$$

Full estimation of the parameters of all $N$ reference sequences across all $M$ serial amplification experiments is given by the solution to the least-squares problem of the sparse, log-linearized system of equations described by the PCR model, following

$$\log x_i(c_j) = \log x_i(0) + c_j \cdot \log \epsilon_i, \forall i \in [1, ..., N], \forall j \in [1, ..., M]. \tag{2}$$

The estimated parameters were finally normalized to their mean. To validate the chosen approach, artificial sequencing data was generated in silico using different defined distributions of initial synthesis bias and relative amplification efficiency. In a first test, a simple model based on Eq.( 1) was used to generate sequencing data without the stochastic effects of PCR, dilution, and sequencing. In a second test, the full workflow was implemented in a digital twin of the DNA data storage process[20] to investigate the approach's robustness against stochastic noise. These validations of the model showed sufficient reliability of the parameter estimates under the expected experimental noise (e.g., stochastic sampling, or stochastic PCR effects) and no sensitivity to the underlying distribution of the parameters (see Supplementary Fig. 26). Consequently, it was chosen over a more complex statistical model (see Supplementary Note 3 and Supplementary Fig. 19). However, the accuracy of the parameter estimates decreased if a sequence was observed only in a few sequencing runs (Supplementary Fig. 26). Thus, we did not consider sequences which occurred less than two times across a set of sequencing data in the following analysis.

## Efficiency measurements by qPCR

Experimental quantification of the qPCR efficiency was performed for four arbitrarily selected sequences of the GCall oligonucleotide pool, shown in Supplementary Table 6. Of the four selected sequences, #00006 had an estimated amplification efficiency of 0.999, #01634 of 1.014, #11493 of 0.854, and the efficiency of #09807 could not be estimated because it was filtered out due to occurring in only one sequencing run (relative coverage after first amplification: 0.21, thereafter 0). Sequence #09807 was included nonetheless to assess whether such sequences were missing in the sequencing data due to stochastic effects or because of an extremely poor amplification efficiency. These oligonucleotide sequences were synthesized individually by Microsynth (Balgach, Switzerland), dissolved with ultrapure water to 100 μM, serially diluted five times by factors of 10x, and each dilution measured by qPCR with the standard primers (0 F/0 R) in duplicates to create a calibration curve. A total of three independent dilutions and qPCR runs were performed for each oligonucleotide sequence, with the results shown in Supplementary Fig. 2 and Supplementary Table 2. The qPCR efficiency of the GCall oligonucleotide pool itself was also measured in triplicates with the same range of dilutions. A comparison of qPCR-derived efficiencies was performed using one-way independent ANOVA after testing for homoscedasticity with Levene's test, followed by post-hoc testing using Tukey's range test.

An additional qPCR-based efficiency measurement of the four sequences (#00006, #01634, #09807, and #11493) with 5′-degenerate

primers (four W at the 5′-end) was performed as above. The results of the calibration curves are shown in Supplementary Fig. 3.

## Selection of literature datasets

Multiple additional sequencing datasets from the literature were selected to test and train the classification model. For parameter estimation, datasets must include sequencing data for at least two different PCR cycle counts, and their sequencing coverage must be sufficiently high to yield accurate coverage distributions. Moreover, to preclude any possible bias stemming from sequences with biological function or extreme sequence properties (such as GC content or long nucleotide repeats), we limited our search to datasets derived from synthetic oligonucleotide pools which contain close-to-random sequences. Multiple such datasets were identified in the DNA data storage literature, from Erlich et al.[22], Koch et al.[49], Song et al.[50], Choi et al.[18], and Gao et al.[19], and processed as described above. A detailed overview of the experimental parameters and sequencing endpoints of all literature datasets is given in Supplementary Table 7.

## Machine learning models

In this work, the main model we propose to use is a 1D-CNN model to predict whether a DNA sequence is of low PCR amplification efficiency given the sequence data. Due to the absence of prior knowledge regarding the specific threshold for low PCR efficiency in randomly synthesized sequences, we empirically determine a 2% threshold to differentiate between low-efficiency sequences and normal-efficiency sequences (see above). This binary categorization then facilitates the formulation of the prediction task as a classification problem.

1D-CNN models have demonstrated superior efficacy compared to traditional machine learning methods across a wide range of DNA sequence property prediction tasks, as discussed in the introduction. While 1D-CNNs can implicitly capture sequential order through multiple layers, they lack explicit modeling of the nucleotide positions. This explicit positional information could be crucial for identifying motifs, especially in shallow models where the implicit capture of order might not be sufficient. For example, in eukaryotic chromosomes, telomeres are repetitive nucleotide sequences at each end of a chromosome. The specific motif sequence often comprises repetitions of a short DNA sequence (like *TTAGGG* in vertebrates). The positional specificity of these motifs at the very ends of the chromosomes is vital for their function in protecting the chromosome from deterioration or fusion with neighboring chromosomes[60,61]. Motivated by this, an additional positional encoding (PE) component, which was first introduced with the transformer model[62], is incorporated into the 1D-CNN model.

Specifically, positional encoding represents nucleotide positions as real-valued vectors, which can then be added to the embeddings of nucleotides. It is defined as:

$$\text{PE}(p, 2i) = \sin\left(\frac{p}{10000^{\frac{2i}{d}}}\right),$$
$$\text{PE}(p, 2i+1) = \cos\left(\frac{p}{10000^{\frac{2i}{d}}}\right),$$

where sinusoidal functions are used to encode each position $p$ into a vector of dimension $d$ (assumed to be an even number), and $i \in [0, d/2]$ represents the dimension index as used in Vaswani et al.[62]. In our implementation, the one-hot encoded DNA sequence is first projected into a higher dimensional space (of the same dimension as PE's dimension). Then, we perform an element-wise addition between the PE and the high-dimensional projection of the sequence.

## Baseline models

To demonstrate the efficacy of the 1D-CNN model in identifying sequences with lower PCR efficiency, we established several baseline

**Table 1 | Hyperparameter grid and ranges for the hyperparameter search of the 1D-CNN model**

| Hyperparameter | Search values |
|---|---|
| Number of convolutional layers | 1, 2, 3 |
| Number of convolutional filters | 32, 64, 128 |
| Length of convolutional filters | 4, 8, 12 |
| Learning rate | $10^{-3}, 10^{-4}, 10^{-5}$ |
| Global pooling | mean, max |
| Batch size | 64, 128, 256 |
| Weight decay | $0, 10^{-3}, 10^{-4}$ |

models for comparative evaluation of the proposed 1D-CNN model with PE. The baseline models include:

- 1D CNN model without PE.
- Recurrent neural network (RNN)-based model.
- LightGBM model with *k*-mer feature encoded with position information.
- Lasso regularized logistic regression (LR) model with the frequency of each nucleotide and the GC content in the sequence.

More details of the baseline models and the architecture of our proposed model can be found in Supplementary Note 1 and 2.

### Experimental setup

Given the binary classification nature of the task, we minimize the binary cross-entropy loss during training. We use two metrics for our analysis: the Area Under the Receiver Operating Characteristic (AUROC) and the Area Under the Precision-Recall Curve (AUPRC). We consider two evaluations: the standard within-dataset evaluation and external validation. For within-dataset evaluation, we employ stratified 5-fold nested cross-validation (CV) to preserve the percentage of samples for each class across folds. In each fold, 10% of the training data is selected as the validation set, which is used only for hyperparameter selection. We perform a randomized search[63] over 50 iterations, each corresponding to a hyperparameter configuration including the learning rate, width, depth, batch size, weight decay, dropout, etc., detailed in Table 1 with an equivalent search conducted for the RNN-based model as described in Supplementary Table 1. The best-performing model parameter configuration is determined by the highest AUPRC over all 5 validation sets and we report the mean and standard deviation of the performance on the 5 test sets. To assess the generalizability of the model, the best-performing hyperparameters from within-dataset evaluations are used to train the model on the entire dataset, which is subsequently tested on different external datasets. We also use class weights inversely proportional to the frequency ratio of the positive class to the negative class, enabling the loss function to more effectively balance the contribution of each class during training. The detailed architecture of the model is illustrated in Fig. 3. The models were implemented in Python (v3.9.7), using numpy (v1.26.4), pandas (v2.2.3), matplotlib (v3.9.4), torch (v2.0.1), torchmetrics (v0.7.2), scikit-learn (v1.2.2), seaborn (v0.12.2), logomaker (v0.8), captum (v0.6.0), scipy (v1.10.1), pytorch-lightning (v1.5.9), and plotly (v5.20.0).

### *CluMo*: motif discovery via attribution and clustering

DNA motifs are short and recurring subsequences found within DNA sequences, which are believed to play critical biological roles. In this study, we introduce CluMo, a method for discovering functional motifs within sequences. Unlike traditional motif discovery tools, CluMo specifically targets motifs associated with user-defined functions, such as PCR amplification efficiency $\epsilon$. It achieves this by integrating feature attribution techniques with k-mer analysis and

clustering. This general approach is applicable to any biological sequence and allows researchers to uncover motifs significantly correlated with any sequence property or function predicted by a deep learning model. We will now delve into the details of CluMo.

### Step 1: feature attribution analysis

CluMo begins by applying a feature attribution method to interpret the prediction made by the trained deep model. In this study, we use DeepLIFT (Deep Learning Important FeaTures)[40], a method that assigns attribution scores to each nucleotide by comparing the activation of neurons to reference activation. Note that our approach is not limited to DeepLIFT and can leverage any attribution analysis method as a replacement, such as Integrated Gradients[64] and SHAP[41]. Using the chosen feature attribution method, nucleotide-level attribution scores are obtained, indicating the impact of each individual nucleotide on the model prediction of each DNA sequence.

### Step 2: significant k-mers identification

To identify the sequence segments with the strongest influence on the model's prediction, we utilize a sliding window approach. We iterate through the sequence with window sizes ranging from 4 to 12 nucleotides (k-mers). For each window size and sequence position, we calculate a cumulative attribution score. This score is the sum of the individual attribution scores (obtained in the previous step) assigned by the deep learning model to each nucleotide within the window. The k-mer with the highest cumulative score, for a given window size, represents the subsequence with the most significant impact on the model's prediction for that specific region of the sequence. These k-mers are selected as the candidates for the next step.

### Step 3: k-mers clustering

We then analyze the relationships between the identified k-mers. To achieve this, we calculate the Hamming distance between each pair of k-mers for each window size. The Hamming distance simply counts the number of nucleotide positions that differ between two k-mers. Next, we employ t-SNE (t-distributed Stochastic Neighbor Embedding)[65] to embed the k-mers into a lower-dimensional (2D) space. This allows us to visualize the relationships between k-mers based on their sequence similarity. Next, we utilize weighted k-means clustering[66] to group similar k-mers. This approach assigns greater weight to more frequent k-mers, ensuring that prevalent sequence patterns exert a stronger influence on the formation of clusters.

Since we lack prior knowledge about the number of functional motifs present, we determine the optimal cluster number $C$ using the silhouette score[67] within the t-SNE embedded space. This score measures the cohesion within clusters and the separation between them. We systematically test values of $C$ from 2 to 12 and select the one that maximizes the average silhouette score across all data points. Finally, for each cluster, we construct a Positional Weight Matrix (PWM) that captures the k-mer frequency within the cluster. This PWM essentially represents the motif for that cluster. Sequence logos can be generated from these PWMs, providing a visually intuitive representation of the identified motifs.

### Step 4: motif presence and enrichment analysis

To identify occurrences of the discovered motifs (PWMs) within a target sequence, we calculate a presence score. This score represents the strongest match between the PWM and any k-mer window across the entire target sequence. The calculation is as follows:

$$\mathcal{F} = \max\left(\frac{\langle P, W_i \rangle}{k}\right) \quad \text{for} \quad i = 1, 2, \ldots, L-k+1,$$

where $P$ denotes the PWM representing the motif, $W_i$ represents the corresponding window in the sequence, $k$ is the window size, and $L$ is

the length of the target sequence. Essentially, the presence score is the maximum dot product between the PWM and each possible k-mer window in the target sequence, normalized by the window size. A score of 1 indicates a perfect match between the PWM and a k-mer in the sequence, while scores below 1 reflect varying degrees of mismatch.

We define a presence threshold of 0.5 for the score $\mathcal{F}$. If the score for a motif in a given sequence exceeds this threshold, we consider the motif to be present. Otherwise, it is considered absent. Following this definition, we can analyze the association between motifs and a specific sequence property. Here, we focus on sequences with low PCR amplification efficiency (positive set). We create the contingency table summarizing the presence/absence of each motif across the test sequences.

Next, we employ a chi-squared test ($\chi^2$ test) to assess the statistical significance of each motif's association with the positive set. The null hypothesis for this test is that there is no connection between the presence of a motif and a sequence belonging to the low PCR efficiency group. The chi-squared statistic is calculated by comparing the observed frequencies of motif presence/absence in the contingency table with the expected frequencies under the null hypothesis. A statistically significant chi-squared value ($p$ value $< \alpha$ with $\alpha = 0.05$) indicates a rejection of the null hypothesis. This suggests a non-random association between the motif and the positive set, implying that the motif might influence PCR efficiency. To account for multiple testing when analyzing numerous motifs, we apply the Bonferroni correction. This adjusts the significance threshold to control for the increased chance of false positives.

### Step 5: motif substitution analysis

To evaluate the influence of each identified motif on the model's prediction, we employ a motif substitution strategy. We substitute the discovered motifs (ranked by their p-value, with the most significant ones first) within the test set sequences. This substitution involves replacing each motif occurrence with a random sequence segment with equal probabilities for each nucleotide.

This substitution process is performed solely on the test set, allowing us to directly compare the model's performance on the original sequences versus the sequences with each motif substituted. By observing these performance changes, we can quantify the impact of each motif on the model's predictions, both within a single dataset and across different datasets.

The detailed algorithm CluMo is described in Algorithm 1.

**Algorithm 1. CluMo.** Motif Discovery via Attribution and Clustering
 **Input:** Low $\epsilon$ sequences $\mathbf{S}$ : $\{s_1, s_2, \ldots, s_n\}$ and the rest sequences $\mathbf{S}'$ : $\{s'_1, s'_2, \ldots, s'_{n'}\}$ each of length $l$; Trained 1D-CNN model $\mathcal{M}$; Feature attribution method (e.g. DeepLIFT)
 **Output:** Motifs contribute to lower $\epsilon$
 **Step 1 and 2 (Feature attribution analysis and significant k-mers identification)**
 **for $k$ = 1 to $n$ do**
 Compute attribution score $z_k$ for sequence $s_k$ using the feature attribution method and $\mathcal{M}$
 **for Window lengths $w$=4 to 12 do**
 Extract most significant subsequence $subseq_{k,w}$ from $z_k$
 **end for**
 **end for**
 **Step 3 (k-mers clustering)**
 **for each window length $w$ = 4 to 12 do**
 Compute pairwise Hamming distance $\mathbf{D}_w$ for all extracted subsequences of length $w$
 Project $\mathbf{D}_w$ to a lower-dimensional representation $\mathbf{D}'_w$ using t-SNE
 Cluster $\mathbf{D}'_w$ using weighted k-means to form clusters $\{C_{1,w}, C_{2,w}, \ldots, C_{m,w}\}$

 **for each cluster $C_{i,w}$ do**
 Compute PWM $P_{i,w}$ for all subsequences in cluster $C_{i,w}$
 **end for**
 **end for**
 **Step 4 (Motif presence and enrichment analysis)**
 **for each PWM $P_{i,w}$ do**
 Compute presence scores of $P_{i,w}$ in all sequences in $\mathbf{S}$ and $\mathbf{S}'$.
 Compute p-value for $P_{i,w}$ based on its enrichment in $\mathbf{S}$ compared to in $\mathbf{S}'$ using $\chi^2$ test.
 Adjust p-values for multiple comparisons using Bonferroni correction
 **if $P_{i,w}$ is statistically significantly enriched then**
 **return** $P_{i,w}$ as a discovered motif
 **end if**
 **end for**
 **Step 5 (Motif substitution analysis)**
 **for each statistically significantly enriched PWM $P_{i,w}$ ordered by p-value do**
 Substitute $P_{i,w}$ with an average one-hot encoded subsequence in both $\mathbf{S}$ and $\mathbf{S}'$
 Assess $\mathcal{M}$ performance post substitution
 **end for**

### Filter comparisons

Using the sequencing data of the validation pool, the effect of different filtering strategies on the occurrence of low-coverage sequences and the sequence recovery were tested. To represent the state-of-the-art using sequence constraints[22,23], the randomly generated sequences of the validation pool were filtered for a GC content of 40–60%, a maximum homopolymer length of four, and a free energy higher than $-15$ kcal mol$^{-1}$ (using mfold[45]). Out of the total of 10,000 randomly generated sequences, this filter removed 3,413 sequences. To compare this state-of-the-art filter with the trained 1D-CNN, we implemented an alternative filter which removed the same number of sequences from the full set, but used only the probability estimates for the positive class (i.e., low amplification efficiency) generated by the trained 1D-CNN.

To estimate the required minimum sequencing depth to achieve a certain sequence coverage, the sequencing data generated for the validation pool under the GCall/GCfix condition was downsampled to different sequencing depths. Each downsampling step was repeated for a total of 30 runs, with the mean sequence recovery across all runs being reported.

### Reporting summary

Further information on research design is available in the Nature Portfolio Reporting Summary linked to this article.

## Data availability

The experimental sequencing data generated in this study has been deposited in the European Nucleotide Archive under accession code PRJEB77604. Literature sequencing datasets are available from Gimpel et al.[20] (European Nucleotide Archive, PRJEB65931), Koch et al.[49] (European Nucleotide Archive, PRJEB35217), Erlich et al.[22] (European Nucleotide Archive, PRJEB19305 and PRJEB19307), Song et al.[50] (Figshare, 16727122, 17193128, and 18515045), Gao et al.[19] (pers. communication), and Choi et al.[18] (European Nucleotide Archive, PRJNA555140). Source data are provided with this paper.

## Code availability

The code used to develop the model, perform the analyses and generate results in this study is publicly available and has been deposited in GitHub at github.com/BorgwardtLab/PCR-bias, under a BSD

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

## Acknowledgements

This project was partially financed by the European Union's Horizon 2020 Program, FET-Open: DNA-FAIRYLIGHTS, grant agreement no. 964995, and the European Union's Horizon 2020 Research and Innovation Program under Marie Sklodowska-Curie Grant Agreement No. 813533. Core funding by the Max Planck Society (to K.B.) and ETH Zürich (to W.J.S.) is acknowledged. Data analysis was performed on the Euler cluster operated by the High-Performance Computing group at ETH Zürich. Figures were partially created with BioRender.com (see BioRender. Gimpel, A. (2025) https://BioRender.com/19eu0ya) and both icons in Fig. 1 (conical microtube and thermocycler) as well as the icon of the conical microtube in Fig. 6a were provided by Labicons (www.labicons.net).

## Author contributions

Conceptualization: K.B., R.N.G.; Methodology: K.B., R.N.G., W.J.S., B.C., A.L.G., B.F., D.C., M.H.; Supervision: D.C., K.B., R.N.G.; Resources: W.J.S., K.B., R.N.G.; Investigation: A.L.G., P.A., L.O.D.W.; Software: B.F., D.C., M.H.; Data analysis: A.L.G., B.F., D.C., M.H., L.M., B.C.; Writing-Draft: A.L.G., B.F., D.C., B.C., R.N.G., K.B.; Visualization: A.L.G., B.F., D.C., B.C., R.N.G., K.B.; Writing-Review: all authors.

## Funding

## Competing interests

The authors declare no competing interests.
