## [Transparent Peer review file · Nature Communications]

Predicting sequence-specific amplification efficiency in multi-template PCR with deep learning

Corresponding Author: Professor Robert Grass

Version 0:

Reviewer comments:

Reviewer #1

(Remarks to the Author)

The study develops a deep learning-based approach to predict and optimize sequence-specific amplification efficiency in multi-template PCR, enabling uniform amplification of amplicons and significantly reducing the required sequencing depth.

The work is extensive and robust, and the article is easy to follow. It has potential to make a significant contribution to the state of the art; however, there are issues that require attention.

The model shows good overall classification performance with an AUROC of 0.88, indicating strong discrimination between classes. However, its AUPRC of 0.44 suggests it struggles with correctly identifying positive instances, likely due to class imbalance. It is recommended to address this imbalance by exploring techniques such as over-sampling or under-sampling, adjusting the decision threshold, or improving the model's focus on the positive class to enhance precision and recall for the minority class.

Are the 12,000 sequences truly "random" in the sense of having equal nucleotide probabilities? In a genuinely random sequence, each nucleotide should have an equal probability of appearing, assuming the design specifies uniform likelihood for all symbols. Over sufficiently long sequences, the nucleotide distribution should approximate uniformity, meaning each nucleotide appears at roughly the same frequency. In this dataset, "random" sequences are 108 nucleotides long, and the authors filtered for GC content either higher than 60% or lower than 40%. While some deviations from uniformity can be expected due to the relatively short sequence length, achieving such extreme GC content values would be challenging if the random generator produces unbiased symbols. This raises questions about the randomness of the sequences. What random number generator was used in this study? Was it a Linear Congruential Generator (LCG), or another method? Please, specify.

Why were only "random" sequences considered? Wouldn't introducing deliberate redundancy in some sequences provide valuable insights into its implications on the results? Examining both random and intentionally biased sequences could offer a more comprehensive understanding of how sequence variability impacts the study's outcomes. For example, low-complexity regions (LCRs) can impact the study by introducing amplification bias, reducing predictive accuracy, and causing non-specific binding, which compromises the uniformity and reliability of results. Additional experiments, removing or manipulating the LCRs, could also confirm whether non-homogeneity is associated with these regions.

Regarding the CluMo software (<https://github.com/BorgwardtLab/PCR-bias>), could the authors please specify the license under which it is distributed? Additionally, it would be helpful to include a simple demo in the README file, along with a more detailed description of its functionality, dependencies, and usage instructions. This would greatly assist users in understanding how to effectively utilize the software. Additionally, consider adding the software to automated installation platforms, such as Bioconda, which can facilitate easier distribution and increase its visibility within the scientific community.

Equation 1 and 2 are currently presented in isolation, lacking contextual integration into the surrounding text. Consider formatting it more like inline text, including appropriate punctuation, and introducing it explicitly before its appearance. For guidance, see this example: <https://i.stack.imgur.com/l4p15.png>

Supplementary Notes contain equations that also require attention.

(Remarks on code availability)

Feedback on CluMo Software (<https://github.com/BorgwardtLab/PCR-bias>):

Licensing: Could the authors please specify the license under which CluMo is distributed? Clearly stating the license will help users understand the terms of use and potential for contribution.

Documentation Improvements: It would be highly beneficial to include a simple demo in the README file, along with a comprehensive description of the software's functionality, dependencies, and detailed usage instructions. This will greatly assist users in effectively understanding and utilizing the software.

Distribution: Consider adding the software to automated installation platforms, such as Bioconda. This would simplify the installation process for users and enhance the software's visibility within the scientific community.

Installation and Usage Issues:

After cloning the repository, I encountered the following error when attempting to run the software:

```
python CluMo.py
Traceback (most recent call last):
  File "/home/user/git/PCR-bias/CluMo.py", line 1, in <module>
    from captum.attr import DeepLift
ModuleNotFoundError: No module named 'captum'
```

To resolve this, I attempted the following command:

```
conda install captum -c conda-forge
```

After resolving the initial issue, I encountered a subsequent error upon re-running the software:

```
python CluMo.py
ModuleNotFoundError: No module named 'utils.utils'
```

Could you please provide detailed installation instructions, including all required dependencies, to help users avoid these issues? This would significantly improve the user experience and ease of adoption.

Reviewer #2

(Remarks to the Author)

This is a solid paper asking how the sequence of a PCR template influences its amplification efficiency. The setup of the experiment is a pool of synthetic sequences that all share the same primer binding sites. The question then is how the varying sequence at the center of the oligo influences PCR amplification efficiency. The authors observe that within a large pool there are two sub-populations. A first population that amplifies well and a second, smaller population that amplifies less efficiently. The authors then train a CNN classifier that can assign sequences to either class with good accuracy. Model interpretation finds a sequence motif that is enriched in the low-performing sequences. This motif can form hairpins with the constant primer site, suppressing primer hybridization and amplification.

My enthusiasm is diminished because the major observations in the paper are extremely specific to the set of primers used. Namely, the authors find that a sequence element in the primer binding sequence can create a hairpin with complementary GC-rich elements when they occur near the 5' end of the varying template sequence. This "adapter-template self priming" competes with primer binding and reduces amplification efficiency. Moreover, the paper also claims to introduce a novel CNN interpretation method and I find the claims about this method to be vastly exaggerated.

Major issues:

1. In the introduction the authors write "Subsequently, DeepLIFT40 and SHAP41 further improved feature attribution analysis for general deep learning models, providing nucleotide-level attribution scores based on specific tasks. However, a key limitation of these methods lies in their focus on individual sequences, often neglecting a comprehensive overview of common motifs across the entire dataset. Their utility is therefore limited for global model explanation and general motif discovery, which would be crucial to better understand effects across multiple sequences, as is required for understanding the nature of inhomogeneous amplification during multi-template PCR." The claims made about the novelty of the current interpretation method (e.g. paragraph above and the corresponding paragraph in the discussion) are misleading. There is a vast literature in genomics that shows how common sequence motifs can be derived from SHAP scores and similar through clustering. These methods are reviewed for example in G Novakovsky et al. Nature Reviews Genetics 24 (2), 125-137".

2. It would be very interesting to discuss (or maybe even do some experiments) on generalizing the current work. One obvious generalization would be to switch the primer sequences and then see whether the same mechanism occurs but now selecting for a different sequence motif (i.e. one complementary to a motif in the new primers). A more challenging but also more general problem would be multiplexed PCR where the primer sequences themselves are varied.

3. The features used in the regression model are very simple. I don't doubt that a CNN will still do better, but I would recommend comparing the CNN to a more expressive regression model, that uses k-mer counts as features. Even better, use k-mer counts with position information (either using k-mer counts at each position or breaking the sequence into regions).

4. Why train a classifier rather than trying to predict the actual amplification efficiency for each sequence? Although it does seem like there are two major classes of sequences, i.e. a subset that does not amplify well and larger set that does amplify, the efficiency measurements are actually finer-grained than that. It wouldn't be surprising if the training a model to predict efficiency directly would result in an overall better model. Of course this is only true if the actual amplification efficiencies are quantitatively reproducible and not just the class labels.

5. Related to the previous point and Fig. 2D,E: It is valuable to test the amplification efficiency for different sequences in the pool using different initial amounts to show that efficiency is an intrinsic property of the sequence. It would be nice to show examples of individual sequences in addition to showing the overall distribution. E.g. show the sequencing read count as a function of the PCR cycles for a few sequences as sketched in Fig. 1A (panel 4) but using real data. Include "replicates" where the initial amounts of the template are varied. Such experiments could presumably also be performed using individual column-synthesized template DNA if the claim is that that the amplification efficiency is an intrinsic property of the sequence (as done in Fig. 2D).

6. Can you perform a model-free analysis of the impact of different 4-mers? Or even 4-mers with position? This could be done by comparing the average amplification efficiency of sequence containing a specific 4-mer to sequences that do not.

7. What fraction of sequencing reads cannot be mapped to a unique template, e.g. because they are primer dimers or something else? Is there any information in these unexpected reads that might help further refine the model?

8. How does amplification efficiency change with oligo length? Can the model generalize to templates of different length?

(Remarks on code availability)

Reviewer #3

(Remarks to the Author)

My comments are in the attached PDF

(Remarks on code availability)

Version 1:

Reviewer comments:

Reviewer #1

(Remarks to the Author)

The authors have addressed my previous concerns and suggestions. However, there are still some issues with the computational tool that need attention.

During installation, I encountered compatibility issues due to the requirement for a specific Python version. This requirement should be clearly stated in the README file at the beginning, to avoid confusion and installation errors.

After resolving the installation issues, I was able to run the tool.

```
$ python CluMo.py
usage: CluMo.py [-h] --filename FILENAME [--threshold THRESHOLD]
CluMo.py: error: the following arguments are required: --filename
```

However, I encountered the following problems when testing the demo:

```
$ python CluMo.py --filename dataset
Traceback (most recent call last):
File "/home/x/git/review_test/PCR-bias/CluMo.py", line 193, in <module>
motif_analysis.motif_plot()
File "/home/x/git/review_test/PCR-bias/CluMo.py", line 134, in motif_plot
p_values_per_pwm = pd.read_pickle(
File "/home/x/git/review_test/PCR-bias/myenv/lib/python3.9/site-packages/pandas/io/pickle.py", line 185, in read_pickle
with get_handle(
```

File "/home/x/git/review_test/PCR-bias/myenv/lib/python3.9/site-packages/pandas/io/common.py", line 882, in get_handle
handle = open(handle, ioargs.mode)

FileNotFoundError: [Errno 2] No such file or directory: 'CNN/motifs/dataset/2perc//significantly_enriched_motifs.pkl'

There is no file inside: "CNN/motifs/dataset/2perc/" folder.

Please, correct this and try in a new machine so you can follow the process without errors.

Additionally, consider to use LLMs to improve the README presentation with proper verification at the end.

(Remarks on code availability)

Reviewer #2

(Remarks to the Author)

The reviewers have largely addressed my comments and the paper can be published once the authors address two minor points:

1. The claims about the interpretation method still seem a bit exaggerated given the state of the art. If you want to claim this as a general purpose method that does better than state-of-the art approaches, it would be best to apply it to other deep learning models (e.g. from the genomics field) and benchmark performance against other methods.

2. I have a hard time believing that Fig. 29 shows the best-performing regression model that can be trained. If I'm interpreting row C correctly, the model simply predicts a single value independent of the sequence or measured performance. That just looks like something went wrong with training data selection, regularization or something else.

(Remarks on code availability)

Reviewer #3

(Remarks to the Author)

The authors have sufficiently addressed my concerns in their revisions.

(Remarks on code availability)

Version 2:

Reviewer comments:

Reviewer #1

(Remarks to the Author)

Thank you for addressing my concerns.

I was able to run the tool successfully after setting up a conda environment with the required Python version (it might be helpful to include this information in the documentation).

Just a minor suggestion: it would be great if the number of warnings generated by the tool could be reduced.

(Remarks on code availability)

Response to Referees

Reviewer comments in *italics*, author replies in blue with actions **bolded**.

Line numbers, as well as Figure and Table numbers, refer to the revised manuscript, in which changes have been highlighted in yellow.

Comments by Referee #1

Comment 1.0

The study develops a deep learning-based approach to predict and optimize sequence-specific amplification efficiency in multi-template PCR, enabling uniform amplification of amplicons and significantly reducing the required sequencing depth.

The work is extensive and robust, and the article is easy to follow. It has potential to make a significant contribution to the state of the art; however, there are issues that require attention.

We would like to thank the reviewer for their thoughtful comments, valuable feedback, and constructive suggestions, which have helped us to improve and clarify our manuscript and enhance the accessibility of our code considerably.

Comment 1.1

The model shows good overall classification performance with an AUROC of 0.88, indicating strong discrimination between classes. However, its AUPRC of 0.44 suggests it struggles with correctly identifying positive instances, likely due to class imbalance. It is recommended to address this imbalance by exploring techniques such as over-sampling or under-sampling, adjusting the decision threshold, or improving the model's focus on the positive class to enhance precision and recall for the minority class.

We appreciate the reviewer's comment regarding the challenges posed by class imbalance. We would like to clarify that the classification task in our study is performed at a prevalence of 2%. In this context, an AUPRC of 0.44 represents a strong performance, substantially above a random classifier, which is aligned with the prevalence rate (i.e., an expected AUPRC of 0.02 in a random classifier).

To further address the comment about class imbalance, we experimented with oversampling techniques. Specifically, **we applied the SMOTE function from the scikit-learn package to oversample the minority class in both internal and external validations** on the GCfix and GCall datasets. However, as shown in Supplementary Fig. 28 (also shown below), models trained with oversampled data did not outperform the models without oversampling, instead but have suffered from significant performance loss. This result suggests that oversampling might distort the underlying data distribution, which can be detrimental to generalization – an observation also supported by prior work.¹

Based on these findings and considering the high AUPRC relative to the class prevalence, we chose to proceed with models trained on the original imbalanced data, which showed superior performance. Nonetheless, we have **added a more detailed discussion of this additional baseline proposed by the reviewer and shown in Supplementary Fig. 28 to Supplementary Note 2.**

Supplementary Fig. 28 Performance comparison between 1D-CNN model with and without over sampling of the minority class (SMOTE) technique on the internal and external validation of GCfix and GCall datasets.

Comment 1.2

Are the 12,000 sequences truly "random" in the sense of having equal nucleotide probabilities? In a genuinely random sequence, each nucleotide should have an equal probability of appearing, assuming the design specifies uniform likelihood for all symbols. Over sufficiently long sequences, the nucleotide distribution should approximate uniformity, meaning each nucleotide appears at roughly the same frequency. In this dataset, "random" sequences are 108 nucleotides long, and the authors filtered for GC content either higher than 60% or lower than 40%. While some deviations from uniformity can be expected due to the relatively short sequence length, achieving such extreme GC content values would be challenging if the random generator produces unbiased symbols. This raises questions about the randomness of the sequences. What random number generator was used in this study? Was it a Linear Congruential Generator (LCG), or another method? Please, specify.

The sequences generated for the GCall and GCfix oligonucleotide pools were indeed randomly generated assuming equal nucleotide probabilities. Specifically, the function `choices` from Python's `random` library was used to select from the four nucleobases at random, creating a fixed sequence length of 108. Internally, `random.choices` uses the Mersenne Twister as the pseudorandom number generator.

As noted in l. 471, the GCfix pool with a constrained GC content was constrained to sequences with exactly 50% GC content (in contrast to the 40-60% GC mentioned by the reviewer). This was implemented by filtering generated sequence for a GC content of exactly 50%. We agree with the reviewer that purely randomly generated sequences should only rarely deviate from a GC content of 40-60%. As we limited sequences to exactly 50% GC however, the necessity to filter the random output despite the use of an unbiased random number generator is evident.

To clarify this in the manuscript, we have **added a subclause to the Methods that highlights the random number generator used for generating sequences**, see ll. 472-474.

Comment 1.3

Why were only "random" sequences considered? Wouldn't introducing deliberate redundancy in some sequences provide valuable insights into its implications on the results? Examining both random and intentionally biased sequences could offer a more comprehensive understanding of how sequence variability impacts the study's outcomes. For example, low-complexity regions (LCRs) can impact the study by introducing amplification bias, reducing predictive accuracy, and causing non-specific binding, which compromises the uniformity and reliability of results. Additional experiments, removing or manipulating the LCRs, could also confirm whether non-homogeneity is associated with these regions.

We agree with the reviewer that non-random sequences would provide additional insights into the sources of amplification bias. Indeed, our validation experiment outlined in Fig. 6 already demonstrated how examining random and intentionally biased sequences in parallel supports the assessment of sequence variability's impact on our results. Specifically, the introduction of specific motifs at different positions and, importantly, in multiple sequences, demonstrated the reliability of the motif-induced effects. As shown in Supplementary Fig. 9, comparing the original sequences' amplification efficiencies to those of sequences with deliberately introduced motifs shows a consistent negative effect on amplification efficiency across 40 different sequences. Moreover, it showcases the diminished effect and larger variation for motifs further downstream from the 5'-adapter.

Nonetheless, we believe it is important to stress that the framework presented in this manuscript is intended for a hypothesis-free sequence analysis. The use of non-random sequences – as proposed by the reviewer – would fall short of this aspiration, as it biases the training data towards the effects anticipated by the sequence design. Of course, we believe the use of deliberately introduced sequence features would prove immensely useful to further explore and validate their effects on amplification efficiency, as we already started in our validation experiments of Fig. 6. However, as discussed above, we do not consider this the primary motivation for our study, but plan to explore this in the future. **To clarify our intention to the reader we have added clarifications to ll. 118-120 of the manuscript.**

Comment 1.4

Regarding the CluMo software (<https://github.com/BorgwardtLab/PCR-bias>), could the authors please specify the license under which it is distributed? Additionally, it would be helpful to include a simple demo in the README file, along with a more detailed description of its functionality, dependencies, and usage instructions. This would greatly assist users in understanding how to effectively utilize the software. Additionally, consider adding the software to automated installation platforms, such as Bioconda, which can facilitate easier distribution and increase its visibility within the scientific community.

We thank the reviewer for their suggestions regarding our CluMo software. As the reviewer has raised these suggestions in more detail in other comments below, we have chosen to address the suggestions and provide our responses directly to Comments 1.6 and 1.7.

Comment 1.5

Equation 1 and 2 are currently presented in isolation, lacking contextual integration into the surrounding text. Consider formatting it more like inline text, including appropriate punctuation, and introducing it explicitly before its appearance. For guidance, see this example: <https://i.stack.imgur.com/l4p15.png>
Supplementary Notes contain equations that also require attention.

Thanks for pointing out the equations' isolation and the associated difficulty of understanding their context. They have now been **better integrated into the text**, both in the main text and the Supplementary Notes, following the guidance provided by the reviewer.

Comment 1.6

Feedback on CluMo Software (<https://github.com/BorgwardtLab/PCR-bias>):

Licensing: Could the authors please specify the license under which CluMo is distributed? Clearly stating the license will help users understand the terms of use and potential for contribution.

Documentation Improvements: It would be highly beneficial to include a simple demo in the README file, along with a comprehensive description of the software's functionality, dependencies, and detailed usage instructions. This will greatly assist users in effectively understanding and utilizing the software.

Distribution: Consider adding the software to automated installation platforms, such as Bioconda. This would simplify the installation process for users and enhance the software's visibility within the scientific community.

We thank the reviewer for their valuable suggestions regarding the CluMo software. In response:

- We have **added a BSD-3-Clause License** to the GitHub repository to clarify the terms of distribution and usage.
- To improve usability, we have **expanded the README file to include quick demos for the main scripts**, along with a more detailed description of the tool's functionality, dependencies, and usage instructions.
- Furthermore, to facilitate easier installation and broader accessibility, we have **published CluMo on PyPI**. Users can now install the package using a simple `pip install clumo` command.

We hope these updates will significantly improve the user experience and facilitate broader adoption of the tool.

Comment 1.7

Installation and Usage Issues:

After cloning the repository, I encountered the following error when attempting to run the software:

```
python CluMo.py
```

```
Traceback (most recent call last):
```

```
File "/home/user/git/PCR-bias/CluMo.py", line 1, in <module>
```

```
from captum.attr import DeepLift
```

```
ModuleNotFoundError: No module named 'captum'
```

To resolve this, I attempted the following command:

```
conda install captum -c conda-forge
```

After resolving the initial issue, I encountered a subsequent error upon re-running the software:

```
python CluMo.py
```

```
ModuleNotFoundError: No module named 'utils.utils'
```

Could you please provide detailed installation instructions, including all required dependencies, to help users avoid these issues? This would significantly improve the user experience and ease of adoption.

We thank the reviewer for bringing this to our attention. We have updated the GitHub repository to improve usability and reproducibility. Specifically:

- We have **uploaded a requirements.txt file that includes all required dependencies** for running the software.
- We have **clarified the expected working directory and provided example commands** to run the scripts correctly.
- Additionally, we have **included example usages and more detailed instructions** on how to run the different scripts to guide users through typical workflows with the software.

We hope these updates will significantly improve the user experience and facilitate broader adoption of the tool.

Comments by Referee #2

Comment 2.0a

This is a solid paper asking how the sequence of a PCR template influences its amplification efficiency. The setup of the experiment is a pool of synthetic sequences that all share the same primer binding sites. The question then is how the varying sequence at the center of the oligo influences PCR amplification efficiency. The authors observe that within a large pool there are two sub-populations. A first population that amplifies well and a second, smaller population that amplifies less efficiently. The authors then train a CNN classifier that can assign sequences to either class with good accuracy. Model interpretation finds a sequence motif that is enriched in the low-performing sequences. This motif can form hairpins with the constant primer site, suppressing primer hybridization and amplification.

We thank the reviewer for the constructive criticism of our work, including their suggestions for adequate follow-up experiments. We believe our additional, experimental results strengthen our conclusions considerably, and demonstrate their generalizability. We would like to address the other general points raised by the reviewer individually below, before responding to their itemized comments thereafter.

Comment 2.0b

My enthusiasm is diminished because the major observations in the paper are extremely specific to the set of primers used.

Our new experiments now demonstrate the generalizability of our findings by showing identical effects are observed for a completely different primer set selected by Primer3 for a random sequence. Please refer to our answer to Comment 2.2 for details on details on the experiments with other primers.

Comment 2.0c

Namely, the authors find that a sequence element in the primer binding sequence can create a hairpin with complementary GC-rich elements when they occur near the 5' end of the varying template sequence. This

“adapter-remplate self priming” competes with primer binding and reduces amplification efficiency.

Our findings are not limited to GC-rich elements, although the observed effect is expectedly more pronounced for GC-rich elements due to their stability. For example, the largest drop in amplification efficiency in our validation experiment was observed for the TCGTGT motif (see Fig. 6c), which has 50% GC content.

Comment 2.0d

Moreover, the paper also claims to introduce a novel CNN interpretation method and I find the claims about this method to be vastly exaggerated.

We have clarified the claims about our method, highlighting the domains in which our method provides novelty to sequence analysis tasks more specifically. Please refer to our answer to Comment 2.1 for details.

Comment 2.1

Major issues:

1. *In the introduction the authors write “Subsequently, DeepLIFT and SHAP further improved feature attribution analysis for general deep learning models, providing nucleotide-level attribution scores based on specific tasks. However, a key limitation of these methods lies in their focus on individual sequences, often neglecting a comprehensive overview of common motifs across the entire dataset. Their utility is therefore limited for global model explanation and general motif discovery, which would be crucial to better understand effects across multiple sequences, as is required for understanding the nature of inhomogeneous amplification during multi-template PCR. “ The claims made about the novelty of the current interpretation method (e.g. paragraph above and the corresponding paragraph in the discussion) are misleading. There is a vast literature in genomics that shows how common sequence motifs can be derived from SHAP scores and similar through clustering. These methods are reviewed for example in G Novakovsky et al. Nature Reviews Genetics 24 (2), 125-137”.*

We agree with the reviewer that clustering and aggregating local explanations to discover shared motifs are established approaches in genomics, as highlighted by Novakovsky *et al.*, Nat. Rev. Genet. 2023 (now referenced in the introduction as Ref. 43). Similarly, prior literature already extracted global insights from local attribution methods such as SHAP. To clarify how CluMo extends the local-to-global motif paradigm, we have **revised our manuscript to clarify how our method builds on, and differs from, these prior efforts by highlighting the challenges CluMo addresses and stressing its specific contributions (ll. 89-94 and ll. 447-455)**. Specifically:

1. **Alignment with Established Work.** We have acknowledged existing methods (e.g., TFMoDisco²) that cluster local attribution maps to discover global motifs, and note the common issues involved—for instance, deciding on the number of clusters, specifying similarity metrics, and handling variable motif length or resolution. In our revision, we explicitly highlight the challenges that CluMo tackles:
 - *Adaptive cluster counts:* We choose the number of clusters by maximizing silhouette scores in a t-SNE embedding, thus avoiding a purely manual or fixed choice.
 - *Weighted clustering:* We weight frequent k-mers more strongly to reduce fragmentation of abundant motifs and help cluster them together.

- *Statistical validation:* Beyond just discovering motifs, we conduct stringent statistical tests to verify the found motifs enrichment in different sequence groups, and employ substitution experiments to confirm the importance of each motif on the deep learning model's prediction.
2. **CluMo's Specific Contributions.** We stress that CluMo is a general pipeline for transforming per-nucleotide attributions into global motif insights. Its combination of flexible k-mer extraction, distance-based embedding (via Hamming distance/t-SNE), silhouette-score-optimized clustering, and motif substitution validation makes it suitable for any domain (genomics, epigenomics, etc.) where local attributions must be distilled into global, functionally testable motifs. In addition, CluMo is designed to handle complexities unique to multi-template PCR, such as focusing on particular adaptor-flanking regions that are prone to hairpin formation or self-priming.

Comment 2.2

2. It would be very interesting to discuss (or maybe even do some experiments) on generalizing the current work. One obvious generalization would be to switch the primer sequences and then see whether the same mechanism occurs but now selecting for a different sequence motif (i.e. one complementary to a motif in the new primers). A more challenging but also more general problem would be multiplexed PCR where the primer sequences themselves are varied.

We thank the reviewer for their great suggestion, which was also raised by reviewer #3 (see Comment 3.2). We have **performed new experiments demonstrating that our results generalize well to a different primer choice**, using completely new primers and amplicons designed with Primer3Plus³ in a qPCR assay. As expected, the motif-free amplicon amplified efficiently ($94.0 \pm 1.9\%$) while amplicons with primer-specific motifs amplified significantly less efficiently (5'-motif: $78.8 \pm 2.1\%$, 3'-motif: $81 \pm 4\%$). Thus, this experiment not only demonstrates the generality of the mechanism using a different set of optimized primers, but also showcases its validity at both the 5'- and 3'-ends of an amplicon. The **results of this new qPCR experiment (details provided in Supplementary Note 5) were added as Supplementary Fig. 31 (added also below for convenience) and are discussed in the main text (ll. 300-312).**

We agree with the reviewer that the analysis of multiplex PCR would be a logical extension of our work. However, as our work investigates amplification biases in multi-template PCR, we have refrained from conducting further experiments in this direction.

Primer3 template without motif

5' - **CGGACGAATTGCGAATGTTT**AGTGTACCGTAAAGATATGGTAT**CGTTTGACAAAGAGCCACCA**-3'

Primer3 template with 5'-motif

5' - **CGGACGAATTGCGAATGTTT****CGTCCG**CCGTAAAGATATGGTAT**CGTTTGACAAAGAGCCACCA**-3'

Primer3 template with 3'-motif

5' - **CGGACGAATTGCGAATGTTT**AGTGTACCGTAAAGATAT**TGGTGG**CGTTTGACAAAGAG**CCACCA**-3'

Supplementary Fig. 31 qPCR experiments demonstrating the motif-dependent inhibitory effect using a different primer set. Using Primer3Plus,³ a primer set and amplicon (bold) were picked from a random 1000 bp sequence. The corresponding amplicon (Primer3 template without motif, top), as well as amplicons with primer-appropriate motifs at the 5'- (Primer3 template with 5'-motif, middle) and 3'-end (Primer3 template with 3'-motif, bottom) were ordered from Microsynth AG (Balgach, Switzerland). Based on qPCR dilution curves, the amplification efficiency of each amplicon was determined in three experimental replicates. Differences between amplicons' amplification efficiencies were assessed after a one-way ANOVA ($N = 3$ per sample, $F(2, 6) = 26.1$, $p = 0.0011$) using pairwise comparisons with Tukey's range test.

Comment 2.3

3. The features used in the regression model are very simple. I don't doubt that a CNN will still do better, but I would recommend comparing the CNN to a more expressive regression model, that uses k -mer counts as features. Even better, use k -mer counts with position information (either using k -mer counts at each position or breaking the sequence into regions).

We appreciate the reviewer's suggestion to compare the CNN model with a more expressive regression model using k -mer features, including positional information. To address this, we **conducted additional analyses in which we represented each DNA sequence by its k -mer composition**. We tested k -mer lengths of $k = 3$, $k = 4$, and $k = 5$, and for each, we constructed a feature vector capturing the frequency of all k -mers present in the sequence. To incorporate position-specific information, we further annotated each k -mer with its start position in the sequence. We note that we did not include $k \geq 6$ in our experiments because the sequences in our dataset are relatively short, and higher-order k -mers are either absent or extremely sparse, limiting their usefulness for training effective models.

We then trained a LightGBM⁴ classifier using these feature representations and compared the performance across different values of k (see Supplementary Fig. 27, reproduced below). Among them, $k = 4$ yielded the highest classification performance. However, even with this optimal configuration, the results were still significantly outperformed by the 1D-CNN model. We have **added a detailed discussion of these k -mer baselines as Supplementary Note 2**, and **added the best-performing 4-mer baseline to the PR and ROC curves in Figure 2 of the manuscript**. Additionally, this baseline is now discussed in ll. 198-201.

Comment 2.4

4. Why train a classifier rather than trying to predict the actual amplification efficiency for each sequence? Although it does seem like there are two major classes of sequences, i.e. a subset that does not amplify well and larger set that does amplify, the efficiency measurements are actually finer-grained than that. It wouldn't be surprising if the training a model to predict efficiency directly would result in an overall better model. Of course this

Supplementary Fig. 27 Performance comparison between k-mer based models and 1D-CNN model on the internal and external validation of GCfix and GCall datasets.

is only true if the actual amplification efficiencies are quantitatively reproducible and not just the class labels.

We thank the reviewer for this thoughtful suggestion. In fact, we initially approached this task using a regression formulation to predict the actual amplification efficiency of each sequence. Thus, we have **repeated our analysis using a regression approach and added implementation details to Supplementary Note 2**. Our analyses revealed that even though the regression model achieved a very low RMSE on average (e.g., 0.008 in internal validation), it struggled to capture meaningful variance in the data (e.g., $R^2 = 0.018$ for GCall and $R^2 = 0.005$ for GCfix) and failed to rank sequences effectively (e.g., rank correlation = -0.014 for GCall and -0.004 for GCfix), particularly within the low-efficiency tail of the distribution. As shown in Supplementary Fig. 29 (reproduced below), the model's predictions are heavily concentrated around the mean.

Further transformation of the regression output into binary probabilities (see Supplementary Note 2) underlined that the regression-derived scores exhibit very limited discriminative power in identifying poorly amplifying sequences (see Supplementary Fig. 29, shown below). All details were **added as Supplementary Fig. 29, with further discussion in Supplementary Note 2 and II. 210-211**.

We note that from an experimental standpoint, the choice of classification over regression is also supported by the larger impact of poor amplification on quantitative applications, as well as the higher reproducibility of poor amplification in our validation experiments.

Supplementary Fig. 29 Performance comparison between regression-driven model and 1D-CNN model on the internal and external validation of GCfix and GCall datasets. The third row shows the direct regression performance, in which the R^2 score, RMSE and rank correlation are presented.

Comment 2.5

5. Related to the previous point and Fig. 2D,E: It is valuable to test the amplification efficiency for different sequences in the pool using different initial amounts to show that efficiency is an intrinsic property of the sequence. It would be nice to show examples of individual sequences in addition to showing the overall distribution. E.g. show the sequencing read count as a function of the PCR cycles for a few sequences as sketched in Fig. 1 (panel 4) but using real data. Include “replicates” where the initial amounts of the template are varied. Such experiments could presumably also be performed using individual column-synthesized template DNA if the claim is that that the amplification efficiency is an intrinsic property of the sequence (as done in Fig. 2D).

The reviewer raises an interesting point about the interplay between initial abundance and amplification efficiency. We would like to point out that Fig. 1a (panel 4) – mentioned by the reviewer – as well as the more detailed illustration in Supplementary Fig. 1, actually show real data (i.e., the GCall dataset). However, these figures do not show examples at different initial concentrations, as suggested by the reviewer. Thus, we have **added additional plots showing the sequencing read count as a function of the PCR cycle**

count, for five sequences with different initial concentrations per group (i.e., poor, normal, and good amplification efficiency). These are provided in Supplementary Fig. 37 (reproduced below).

To conclusively test whether the initial amount affects amplification efficiency, we analyzed the qPCR data using individual column-synthesized template DNA (i.e., Primer3-None, Primer3-5'motif, and Primer3-3'motif), as proposed by the reviewer. Specifically, we checked whether the difference in cycle thresholds between subsequent dilutions was constant – indicating no effect of initial abundance – or declined – indicating a lower abundance increased amplification efficiency – as a function of dilution. As shown in Supplementary Fig. 36 (reproduced below), only a negligible decrease of the difference in cycle thresholds between subsequent dilutions was observed for all templates. This suggests the amplification efficiency remained essentially constant over a concentration range spanning six orders of magnitude. **Supplementary Figs. 36+37 were added to the SI, and a short discussion of this interplay between abundance and efficiency added to ll. 311-312 of the manuscript.**

Supplementary Fig. 37 Exemplary trajectories of the relative coverage of different sequences from the GCall dataset. In each group of poor (left), normal (middle), and good amplification (right), the relative coverage as a function of the number of PCR cycles is shown for five exemplary sequences. Points denote the coverage in the experimental sequencing data, solid lines represent the model fit for estimating initial abundance and amplification efficiency.

Supplementary Fig. 36 Difference in cycle thresholds between subsequent dilutions of the three individually synthesized template sequences Primer3-None (gray), Primer3-5'motif (light red), and Primer3-3'motif (dark red). Shown are the mean of the three individual experimental runs, with brackets indicating the standard deviation. The solid line represents a linear regression, indicating only a minor decrease as a function of dilution.

Comment 2.6

6. Can you perform a model-free analysis of the impact of different 4-mers? Or even 4-mers with position? This could be done by comparing the average amplification efficiency of sequence containing a specific 4-mer to sequences that do not.

We welcome this suggestion by the reviewer. We had already performed a basic, model-free k-mer ($k \in [4, 5, 6]$) analysis in Supplementary Fig. 7 in the initial submission, which provided additional evidence for the relevance of the CGTG submotif. However, we agree with the reviewer that the introduction of positional information into this analysis would further support our conclusions.

For this new k-mer analysis, we have used the same classification of the amplification efficiency as in the rest of our manuscript, and binned the positions of k-mers into five equal-width groups. Then, we assess the log2-enrichment of k-mers in the positive class versus the negative class, adjusted for positive class prevalence (i.e., a value of 0 represents no relative enrichment, > 0 represents enrichment in sequences classified as poorly amplifying). We have limited the analysis to 4-mers and 5-mers, as the additional positional binning reduced the abundance of individual 6-mer/position pairs in the positive class into the single digits.

We have **added this new, extended k-mer analysis with positional information as Supplementary Fig. 6 and discuss it in ll. 240-241 of the manuscript** (shown below for convenience).

Supplementary Fig. 6 It should be noted that the positional binning leads to a low abundance of k-mers within the poorly-amplifying sequences in each group. As a result, the significance of k-mers under-represented in poorly-amplifying sequences is low, as there is considerable stochastic noise.

Comment 2.7

7. What fraction of sequencing reads cannot be mapped to a unique template, e.g. because they are primer dimers or something else? Is there any information in these unexpected reads that might help further refine the model?

The reviewer raises an important point about the post-processing of our sequencing data. In our workflow, i.e., using Illumina sequencing with indexed primers, there are two steps of post-processing that could introduce bias into our analysis. First, the sequencing instrument performs de-multiplexing based on the indexes. Here, reads with erroneous indexes are lost, with the potential for bias if the loss is selective for specific sequences. On the other hand, some reads in the sequencing data cannot be accurately mapped to a reference sequence. Possible

reasons for this include foreign sequences, such as the primer dimers mentioned by the reviewer, or poor quality reads discarded by the mapping software.

To demonstrate that these post-processing steps do not affect our data, we **provide analyses for each of our own datasets in Supplementary Figs. 49-53**. To elucidate the source of unmapped reads further, we **now also directly analysed the raw sequencing data of mapped and unmapped reads with FastQC**. We found that unmapped reads had considerably lower Phred quality scores than mapped reads (≈ 23 vs. ≈ 35), and contained unreasonable amounts of T base calls ($> 40\%$ vs. $\approx 25\%$). Accordingly, we believe the majority of unmapped reads to be caused by poor quality (e.g., truncated) reads from the sequencer being discarded by the mapping software, rather than the presence of foreign sequences such as primer dimers. An **overview of the mean quality score of mapped vs. unmapped reads has been added as Supplementary Fig. 38** (reproduced below).

Supplementary Fig. 38 Mean Phred quality scores of the mapped (blue) and unmapped (red) sequencing reads of the sequencing data for the GCall (left) and GCfix (right) datasets. Shown are the mean quality scores per read, with the standard deviation shown as brackets.

Comment 2.8

8. *How does amplification efficiency change with oligo length? Can the model generalize to templates of different length?*

In the literature, oligo length is known to affect amplification efficiency if the oligo length exceeds the processivity and extension speed of the polymerase.⁵ While our initial data only considered oligos of equal length, our new experiments described for Comment 2.2 used shorter sequences (i.e., 63 nt). There, the same motif-induced effect was observed, while the motif-free template with 63 nt exhibited an amplification efficiency similar to those of templates with 149 nt total. These results thus support the literature findings of negligible impact of length on amplification efficiency, given oligo lengths are sufficiently short to be fully extended within the extension phase. Our 1D-CNN model accepts sequences of arbitrary lengths as its input. **To stress this capability, we have revised l. 450 in the manuscript.**

Comments by Referee #3

Comment 3.0

Gimpel et. al. present a machine learning approach for analyzing PCR bias in multi-template PCRs. After building a CNN, they develop a clustering approach to identify sequence motifs that appear to be drivers of low PCR amplification efficiencies. They observe that these motifs are often close to the adaptor sites and share complementarity with the adaptors and from this observation hypothesize a 3'-adaptor-template self-priming mechanism as a major source of sequence-specific lower amplification efficiencies. The CNN they present serves as a better filter for low PCR amplification efficiency templates than current approaches based on GC and repetitive sequence content. This work has many applications across many fields, including genomics, transcriptomics, and synthetic biology.

The study is technically sound, and the claims are mostly supported by the data presented. I have a few suggestions below that I think would strengthen some of the claims and improve the manuscript.

We thank the reviewer for their useful suggestions. The validation experiments suggested by the reviewer have considerably improved the manuscript by providing additional evidence for our proposed inhibition mechanism.

Comment 3.1

Comments:

1. One additional validation experiment would have been testing a system with different adapter sequences for priming. All experimental results used the 0F / 0R priming sequences. Running a full sequencing experiment is probably not necessary, but a smaller qPCR experiment for some of the alternative adapter sequences / motifs identified in Figure 6 would strengthen the generality claims of the paper.

We agree with the reviewer's comment, and have **performed new qPCR experiments using a different set of primers (see Supplementary Fig. 31)**, as suggested by the reviewer. As reviewer #2 had a similar suggestion in Comment 2.2, we refer to our answer there for details, results, and actions.

Comment 3.2

2. The template hairpin formation mechanism seems plausible, but some more direct evidence for it would strengthen the paper. Would it be possible to include a relatively simple gel electrophoresis experiment to look for the 3' extended template products. Take an individual template sequence that is predicted to have poor amplification efficiency (#09807 for example) and run PCR of a relatively high concentration of template without primers present and then run a gel to see if the extension product is produced at all?

We agree fully with the reviewer's comment. We have **performed new experiments to collect further evidence for the hypothesized hairpin extension**. Specifically, we investigated the occurrence of the extended hairpin structure both during normal amplification (i.e., including primers) and extension of self-primed hairpins only (as suggested by the reviewer, i.e., without primers).

In the primer-free amplification suggested by the reviewer, our hypothesis of hairpin extension predicts the occurrence of a fully complementary hairpin as the only double-stranded product. Indeed, after extension, we observe this extended hairpin during gel electrophoresis (see Supplementary Fig. 35, shown also below). Moreover, the extension of the hairpin is evident from the fluorescence trace (see Supplementary Fig. 33, shown below for convenience) and the changes between pre- and post-extension melting curves (see Supplementary Fig. 34, reproduced below for convenience).

In a normal amplification with primers, our hypothesis of hairpin extension predicts the occurrence of a fully complementary hairpin in parallel to the expected, double-stranded amplicon. These two products should exhibit different melting temperatures, thus we performed amplification and melting curve analyses to also provide evidence for this mechanism during standard PCR conditions. As expected, the amplification with primers also led to the generation of a second species with a higher melting point for the amplicons with motifs (see Supplementary Fig. 32, also shown below).

We have now added **Supplementary Figs. 32-35 to the SI, and provide full experimental details in Supplementary Note 5. Further, we discuss these results as evidence for the hairpin extension in ll. 300-312 of the manuscript.**

Supplementary Fig. 35 Agarose gel electrophoresis (4% with SYBR Gold II, E-Gel EX Agarose Gels, Invitrogen) of the templates picked with Primer3, before ("stock") and after extension ("ext.") without primers. Note that the intensity of samples after extension is lower for the sequence without motif and with the 5'-motif, due to the dilution during extension. In contrast, the sequence with 3'-motif exhibits a higher intensity after extension, due to its double-strandedness after extension. A reference ladder (Ultra Low Range DNA Ladder, Invitrogen) was run in the first and last well (annotated by base pairs on each side). Gel was illuminated on a transilluminator with filter (E-Gel Power Snap Electrophoresis System, Invitrogen) and photographed with a digital camera (Sony RX10II, ZEISS Vario-Sonnar T* F2.8). Final image was cropped, rotated, converted to grayscale and inverted with GIMP 3.

Supplementary Fig. 33 Fluorescence (top) and temperature profiles (bottom) throughout the primer-less extension of the templates picked with Primer3. The template without motif (grey) and with a 5'-motif (blue) did not exhibit a melting transition in both the pre- and post-extension melting curves, and did not increase in fluorescence during the extension phase. In contrast, the template with 3'-motif exhibited a melting transition in the post-extension melting curve and increased in fluorescence throughout the extension phase. Dashed vertical lines denote the boundaries of different measurement phases, and the solid vertical line denotes the addition of a polymerase to each sample. The melting profiles of the pre- and post-extension melting curves are shown separately in Supplementary Fig. 34.

Supplementary Fig. 34 Melting curve analysis of the templates picked with Primer3, before (dashed) and after (solid) extension without primers. The templates without motif (grey) and with 5'-motif (blue) do not exhibit any melting peak both prior and after the extension phase. The template with 3'-motif does not show a melting transition before the extension, but has a distinct melting peak after the extension. Note that this melting curve is not directly comparable to the melting curve shown in Supplementary Fig. 32, as the composition of both master mixes differs (e.g., Mg^{2+} concentration).

Supplementary Fig. 32 Melting curve analysis of the templates picked with Primer3, after normal amplification with primers for 15 cycles. The template without motif (grey) exhibits a singular melting peak, whereas the templates with 5'-motif (blue) and 3'-motif (red) exhibit two melting peaks. Also shown are the predicted melting temperatures of the amplicons (solid lines, via OligoAnalyzer by IDT, eu.idtdna.com/calc/analyzer) and the extended hairpins (dashed lines, via mfold, www.unafold.org/mfold/applications/dna-folding-form.php).

Comment 3.3

3. In the abstract / introduction the authors claim that “CluMo uncovers 3'-adaptor-template self-priming as the major mechanism underlying low sequence-specific amplification efficiency”. This is a bit of stretch because CluMo just identified motifs that were important predictors of amp efficiency, the human authors of this paper then inferred a mechanism for this based on the fact that these important motifs seemed to always be near the adaptor sites. So CluMo is more a guide for researchers to then generate mechanisms / testable hypotheses around.

We agree with the reviewer's observation that CluMo identifies motifs correlated with poor amplification, and that the underlying mechanism – 3'-adaptor-template self-priming – was inferred by the authors based on these motifs' location and behaviour. We have **revised the abstract and l. 108** to clarify that CluMo serves primarily as a guide to identify motifs driving low amplification efficiency, and that our conclusion regarding self-priming is a hypothesized mechanism inspired by those findings rather than a direct discovery by the method itself. Specifically, we **replaced language suggesting that CluMo “uncovers” the mechanism with phrasing that clearly attributes the mechanism to our interpretation of the motifs identified by CluMo.**

Comment 3.4

4. After building the CNN model and using CluMo to formulate a plausible hypothesis of 3'-adaptor-template self-priming, is there still a need for the CNN model / CluMo for future library designs? Couldn't one just use a simple heuristic for filtering now based on not exceeding a certain degree of complementarity with adapter sequences and adjacent template sequences and more or less do as well as the 1D-CNN filtering?

After identifying the mechanism of adaptor-template self-priming, it is true that a simple heuristic based on adapter complementarity could be useful in some cases. However, we believe the 1D-CNN/CluMo approach remains valuable for two main reasons.

First, our model is not limited to this specific issue but offers a hypothesis-free framework applicable to a wide range of sequence analysis tasks, such as identifying hard-to-amplify sequences or correcting amplicon sequencing biases.

Second, even for the specific amplification bias problem discussed, designing an effective heuristic is not straightforward and determining the weighing of motifs correctly may require extensive calibration, whereas the 1D-CNN inherently captures these complexities.

Thus, we believe the model remains relevant both for the current task and offers a useful framework for future research endeavours. **This is now highlighted in ll. 447-448 of the manuscript.**

Comment 3.5

5. For things like RNA-seq where beforehand you know the sequences you are looking for if working with an annotated genome, could the 1D-CNN model be used to identify which sequences might be most susceptible to PCR bias during library prep?

The reviewer raises an important point about extending our 1D-CNN model to other sequence analysis tasks. We are confident that our approach would support the identification of confounding motifs in other applications, provided that training data for the specific application is available. **To highlight the ability of extending our framework to other tasks by training on datasets of other problems, we added a short outlook to ll. 447-448.**

Comment 3.6

6. In the Figure 2 caption can the terms high, medium, and low amplification efficiencies be given more quantitative definitions. I think this same three classes are later referred to as “average, good, or poorly” amplified sequences in the main text, these terms should be unified across the text and the figures.

We thank the reviewer for highlighting the discrepancy in the definitions used throughout our manuscript. We have **harmonized these terms** where applicable, and **added quantitative definitions to these classifications in Fig. 2d+e.**

Comment 3.7

7. In Figure 2b (and some other figure panels later) the figure caption refers to the different colored blue lines in percentages but the figure presents them as fractions. Since the Y-axis of Figure 2b is also percentages I think it is clearer to keep everything in the figure as percentages.

Thanks for pointing out this inconsistency. We have **changed the representation of the fractions in Fig. 2b to percentages.**

Comment 3.8

8. A large number of the Supplementary Figures are not directly referenced in the main text, so it is hard to know which part of the study they go along with.

We have **added additional references to the Supplementary Figures in the main manuscript**, to provide additional context and simplify cross-referencing of the data presented in the SI.

Comment 3.9

Methods:

1. What thermocycler was used for serial dilutions, etc.
2. What qPCR kit was used?
3. Is ultrapure water a product that can be purchased or is this from an inhouse water purification system?

We thank the reviewer for pointing out the lack of these details in our methods. The thermocycler used at ETH Zürich is a LightCycler 480 II (Roche), and at University of Stuttgart a C1000 Touch thermocycler (BioRad) was used. The qPCR kits are KAPA SYBR FAST polymerase master mix from Sigma-Aldrich (St. Louis, MO, United States) and Q5 Hot Start High-Fidelity master mix (Catalog# M0494) from New England Biolabs (Ipswich, MA, United States), as listed in ll. 512 and 516. Both at ETH Zürich and University of Stuttgart, the ultrapure water was from a water purification system (Barnstead MicroPure UV at ETHZ, Barnstead NANOpure II at USTUTT). **These details were added to the methods, at ll. 509+515 and ll. 533-535.**

Comment 3.10

Typos:

1. Page 7 type “As [an] alternative. . . ”
2. Page 17: Sigma-Aldrich (St. Louis, MI, United States) → Sigma-Aldrich (St. Louis, [MO], United States)
3. Page 17: Finally, a final extension at 72 °C for 5 [min] was performed.
4. Page 18 typo: “only in [a] few sequencing runs

We thank the reviewer for pointing out these typos. They have been **corrected in the revised manuscript.**

References

1. Tarawneh, A. S., Hassanat, A. B., Altarawneh, G. A. & Almuhaimeed, A. Stop Oversampling for Class Imbalance Learning: A Review. *IEEE Access* **10**, 47643–47660 (2022).
2. Shrikumar, A. *et al.* Technical note on transcription factor motif discovery from importance scores (TF-MoDISco) version 0.5. 6.5. *arXiv preprint arXiv:1811.00416* (2018).
3. Untergasser, A. *et al.* Primer3Plus, an enhanced web interface to Primer3. *Nucleic Acids Research* **35**, W71–W74 (2007).
4. Ke, G. *et al.* Lightgbm: A highly efficient gradient boosting decision tree. *Advances in neural information processing systems* **30** (2017).
5. Dabney, J. & and, M. M. Length and GC-Biases During Sequencing Library Amplification: A Comparison of Various Polymerase-Buffer Systems with Ancient and Modern DNA Sequencing Libraries. *BioTechniques* **52**, 87–94 (2012).

Response to Referees

Reviewer comments in *italics*, author replies in **blue** with actions **bolded**.

Line numbers, as well as Figure and Table numbers, refer to the revised manuscript, in which changes have been highlighted in **yellow**.

Comments by Referee #1

Comment 1.1

The authors have addressed my previous concerns and suggestions. However, there are still some issues with the computational tool that need attention. During installation, I encountered compatibility issues due to the requirement for a specific Python version. This requirement should be clearly stated in the README file at the beginning, to avoid confusion and installation errors.

After resolving the installation issues, I was able to run the tool.

```
$ python CluMo.py
usage: CluMo.py [-h] --filename FILENAME [--threshold THRESHOLD]
CluMo.py: error: the following arguments are required: --filename
```

We thank the referee again for their careful evaluation of our tool. We acknowledge the oversight in not clearly stating the required Python version. The tool was developed and tested with Python 3.9.7, and we have now **explicitly included this requirement at the top of the README file to avoid any further compatibility issues.**

Comment 1.2

However, I encountered the following problems when testing the demo:

```
$ python CluMo.py --filename dataset
Traceback (most recent call last):
File "/home/x/git/review_test/PCR-bias/CluMo.py", line 193, in <module>
motif_analysis.motif_plot()
File "/home/x/git/review_test/PCR-bias/CluMo.py", line 134, in motif_plot
p_values_per_pwm = pd.read_pickle(
File "/home/x/git/review_test/PCR-bias/myenv/lib/python3.9/site-packages/pandas/
io/pickle.py", line 185, in read_pickle
with get_handle(
File "/home/x/git/review_test/PCR-bias/myenv/lib/python3.9/site-packages/pandas/
io/common.py", line 882, in get_handle
handle = open(handle, ioargs.mode)
FileNotFoundError: [Errno 2] No such file or directory: 'CNN/motifs/dataset/2perc/
/significantly_enriched_motifs.pkl'
```

There is no file inside: "CNN/motifs/dataset/2perc/" folder. Please, correct this and try in a new machine so you can follow the process without errors. Additionally, consider to use LLMs to improve the README

presentation with proper verification at the end.

We sincerely appreciate the referee’s time for testing our tool and highlighting remaining issues. We believe the issue may have stemmed from the placeholder name `dataset` used in the command:

```
python CluMo.py --filename dataset
```

As indicated in the README, the `--filename` argument must be set to one of the 7 actual dataset names used in our study:

1. Choi_et_al
2. Erlich_et_al
3. Gao_et_al
4. GCall
5. GCfix
6. Koch_et_al
7. Song_et_al

Using the literal string "dataset" (as in the example) does not point to a valid input directory, hence the missing file error you encountered. We’ve now **made this requirement even more explicit in the README and added an example using a real dataset name to avoid confusion.**

We’ve also re-tested the tool in a clean environment to confirm that the workflow runs as expected when following the updated instructions.

Comments by Referee #2

Comment 2.1

The reviewers have largely addressed my comments and the paper can be published once the authors address two minor points:

1. *The claims about the interpretation method still seem a bit exaggerated given the state of the art. If you want to claim this as a general purpose method that does better than state-of-the art approaches, it would be best to apply it to other deep learning models (e.g. from the genomics field) and benchmark performance against other methods.*

We understand the referee’s viewpoint, and we agree that we have only applied our interpretation method to the task in this manuscript, rather than performing generalized benchmarks against other methods. Thus, we have now **rephrased the discussion to highlight CluMo’s specific capabilities for this specific problem rather than to compare its capabilities to the state-of-the-art**, i.e. in ll. 447-448:

Our deep learning interpretation framework, CluMo, played a crucial role in enabling the identification of the hairpin formation mechanism.

Comment 2.2

2. *I have a hard time believing that Fig. 29 shows the best-performing regression model that can be trained. If I’m interpreting row C correctly, the model simply predicts a single value independent of the sequence or measured performance. That just looks like something went wrong with training data selection, regularization or something else.*

We appreciate this careful observation and understand the concern. We confirm that the regression models in

Figure 29 were trained using the same pipeline as the classification models. **The corresponding code was now added to the GitHub repository for reproducibility**, see the file `EXP_revision_regression.py` in the top-level directory.

The nearly constant predictions observed are not due to implementation errors or data selection issues, but rather reflect a genuine limitation of the regression formulation in this context. Although the model achieves low RMSE values (e.g., about 0.008), the amplification efficiency values are tightly distributed. Prior work [1] demonstrated that the regression objective is harder to optimize compared to the classification counterpart, leading to bad local optima whose support can be very sparse. As a result, the regression model is prone to collapse toward predicting mean values across the dataset - minimizing MSE but failing to capture meaningful variation or sequence-level ranking. This leads to the observed low R-squared and rank correlation values, consistent with findings from prior studies on similar prediction tasks [1].

These results further support our choice of classification-based objectives, which demonstrates substantially better performance in identifying low-efficiency sequences.

References

1. Stewart, L., Bach, F., Berthet, Q. & Vert, J.-P. Regression as Classification: Influence of Task Formulation on Neural Network Features. *arXiv:2211.05641* (2023).

Response to Referees

Reviewer comments in *italics*, author replies in blue with actions **bolded**.

Comments by Referee #1

Comment 1.1

Thank you for addressing my concerns. I was able to run the tool successfully after setting up a conda environment with the required Python version (it might be helpful to include this information in the documentation). Just a minor suggestion: it would be great if the number of warnings generated by the tool could be reduced.

We thank the referee for testing our tool and are glad that they were successful in running it. **The requirement for Python 3.9.7 is explicitly included in the README file, and the warnings have now been removed.**

Gimpel et. al. present a machine learning approach for analyzing PCR bias in multi-template PCRs. After building a CNN, they develop a clustering approach to identify sequence motifs that appear to be drivers of low PCR amplification efficiencies. They observe that these motifs are often close to the adaptor sites and share complementarity with the adaptors and from this observation hypothesize a 3'-adaptor-template self-priming mechanism as a major source of sequence-specific lower amplification efficiencies. The CNN they present serves as a better filter for low PCR amplification efficiency templates than current approaches based on GC and repetitive sequence content. This work has many applications across many fields, including genomics, transcriptomics, and synthetic biology.

The study is technically sound, and the claims are mostly supported by the data presented. I have a few suggestions below that I think would strengthen some of the claims and improve the manuscript.

Comments:

1. One additional validation experiment would have been testing a system with different adapter sequences for priming. All experimental results used the OF / OR priming sequences. Running a full sequencing experiment is probably not necessary, but a smaller qPCR experiment for some of the alternative adapter sequences / motifs identified in Figure 6 would strengthen the generality claims of the paper.
2. The template hairpin formation mechanism seems plausible, but some more direct evidence for it would strengthen the paper. Would it be possible to include a relatively simple gel electrophoresis experiment to look for the 3' extended template products. Take an individual template sequence that is predicted to have poor amplification efficiency (#09807 for example) and run PCR of a relatively high concentration of template without primers present and then run a gel to see if the extension product is produced at all?
3. In the abstract / introduction the authors claim that "CluMo uncovers 3'-adaptor-template self-priming as the major mechanism underlying low sequence-specific amplification efficiency". This is a bit of stretch because CluMo just identified motifs that were important predictors of amp efficiency, the human authors of this paper then inferred a mechanism for this based on the fact that these important motifs seemed to always be near the adaptor sites. So CluMo is more a guide for researchers to then generate mechanisms / testable hypotheses around.
4. After building the CNN model and using CluMo to formulate a plausible hypothesis of 3'-adaptor-template self-priming, is there still a need for the CNN model / CluMo for future library designs? Couldn't one just use a simple heuristic for filtering now based on not exceeding a certain degree of complementarity with adapter sequences and adjacent template sequences and more or less do as well as the 1D-CNN filtering?
5. For things like RNA-seq where beforehand you know the sequences you are looking for if working with an annotated genome, could the 1D-CNN model be used to identify which sequences might be most susceptible to PCR bias during library prep?
6. In the Figure 2 caption can the terms high, medium, and low amplification efficiencies be given more quantitative definitions. I think this same three classes are later referred to as "average, good, or poorly" amplified sequences in the main text, these terms should be unified across the text and the figures.

7. In Figure 2b (and some other figure panels later) the figure caption refers to the different colored blue lines in percentages but the figure presents them as fractions. Since the Y-axis of Figure 2b is also percentages I think it is clearer to keep everything in the figure as percentages.
8. A large number of the Supplementary Figures are not directly referenced in the main text, so it is hard to know which part of the study they go along with.

Methods:

1. What thermocycler was used for serial dilutions, etc.
2. What qPCR kit was used?
3. Is ultrapure water a product that can be purchased or is this from an inhouse water purification system?

Typos:

1. Page 7 type "As [an] alternative..."
2. Page 17: Sigma-Aldrich (St. Louis, MI, United States) -> Sigma-Aldrich (St. Louis, [MO], United States)
3. Page 17: Finally, a final extension at 72 °C for 5 [min] was performed.
4. Page 18 typo: "only in [a] few sequencing runs"